# Transcriptional heterogeneity of ventricular zone cells in the ganglionic eminences of the mouse forebrain

Dongjin R Lee[1], Christopher Rhodes[1], Apratim Mitra[2], Yajun Zhang[1], Dragan Maric[3], Ryan K Dale[2], Timothy J Petros[1]*

[1]Unit on Cellular and Molecular Neurodevelopment, Eunice Kennedy Shriver National Institute of Child Health and Human Development, National Institutes of Health, Bethesda, United States; [2]Bioinformatics and Scientific Programming Core, Eunice Kennedy Shriver National Institute of Child Health and Human Development, National Institutes of Health, Bethesda, United States; [3]Flow and Imaging Cytometry Core, National Institute of Neurological Disorders and Stroke, National Institutes of Health, Bethesda, United States

**Abstract** The ventricular zone (VZ) of the nervous system contains radial glia cells that were originally considered relatively homogenous in their gene expression, but a detailed characterization of transcriptional diversity in these VZ cells has not been reported. Here, we performed single-cell RNA sequencing to characterize transcriptional heterogeneity of neural progenitors within the VZ and subventricular zone (SVZ) of the ganglionic eminences (GEs), the source of all forebrain GABAergic neurons. By using a transgenic mouse line to enrich for VZ cells, we characterize significant transcriptional heterogeneity, both between GEs and within spatial subdomains of specific GEs. Additionally, we observe differential gene expression between E12.5 and E14.5 VZ cells, which could provide insights into temporal changes in cell fate. Together, our results reveal a previously unknown spatial and temporal genetic diversity of VZ cells in the ventral forebrain that will aid our understanding of initial fate decisions in the forebrain.

*For correspondence:
tim.petros@nih.gov

**Competing interest:** The authors declare that no competing interests exist.

## Editor's evaluation

Your study is a thorough transcriptomics study demonstrating previously uncharacterized transcriptional diversity, discriminating the embryonic domains that produce interneurons. You have responded well to the requests for amendments, adjusted figures and better emphasized gene diversity in the ganglionic eminence.

## Introduction

During embryogenesis, the telencephalon is derived from a neuroepithelium sheet at the anterior end of the neural plate. Along the neuraxis, the telencephalon develops an anterior-posterior and dorsal-ventral identity guided by extrinsic signals (*Wilson and Houart, 2004*). The dorsal telencephalon gives rise to excitatory glutamatergic cells of the cortex (CTX), whereas the ventral telencephalon is comprised of three transient structures, the lateral, medial and caudal ganglionic eminences (LGE, MGE, and CGE, respectively), that give rise to all forebrain GABAergic cells (*Hébert and Fishell, 2008*). Each of these regions contains progenitor domains known as the ventricular zone (VZ), which harbors *Nestin*-expressing radial glia cells (RGCs) and apical progenitors (APs), and the subventricular zone (SVZ) where basal progenitors (BPs) reside (*Batista-Brito et al., 2008*; *Flames*

*et al., 2007*; *Inan et al., 2012*; *Long et al., 2009*; *Petros et al., 2015*; *Rowitch and Kriegstein, 2010*; *Turrero García and Harwell, 2017*; *Wonders et al., 2008*). Early studies supported the hypothesis that RGC progenitors in the cortex become progressively restricted over future cell divisions (*Desai and McConnell, 2000*; *Gao et al., 2014*; *McConnell and Kaznowski, 1991*). More recent experiments identified some heterogeneity of forebrain VZ cells (*Franco and Müller, 2013*; *Llorca et al., 2019*), but a comprehensive characterization of these VZ neural progenitor cells has not been performed.

There is a significant expansion in the number of SVZ cells as embryogenesis progresses which is likely related to developmental changes in cell fate (*Lui et al., 2011*). Several mechanisms have been shown to guide cell fate decisions in the ventral telencephalon, including spatial gradients of signaling factors (*Brandão and Romcy-Pereira, 2015*; *Flames et al., 2007*; *McKinsey et al., 2013*; *Tyson et al., 2015*; *Wonders et al., 2008*; *Xu et al., 2010*), cellular birthdates (*Bandler et al., 2017*; *Inan et al., 2012*; *Miyoshi and Fishell, 2011*; *Rymar and Sadikot, 2007*) and the mode of neurogenesis (*Petros et al., 2015*), all of which influence the cell cycle dynamics of VZ and SVZ cells. Additionally, many disease-associated genes are enriched in RGCs and interneuron progenitors (*Schork et al., 2019*; *Trevino et al., 2020*). Thus, a thorough characterization of gene expression in VZ and SVZ within the ganglionic eminences will increase our understanding of initial cell fate decisions of GABAergic cells in the embryonic brain and provide insight into possible disease mechanisms.

In recent years, single-cell RNA-sequencing (scRNAseq) studies revealed extensive neuronal diversity in the mature telencephalon whereby cells can be classified into relatively clean distinct neuronal subtypes based on their transcriptome (*Economo et al., 2018*; *Tasic et al., 2016*; *Tasic et al., 2018*; *Zeisel et al., 2015*). However, significantly less is known about neural progenitor diversity within the developing embryo where initial fate decisions arise. Several initial studies performed bulk RNA-sequencing on the embryonic ventral forebrain, but these studies were not able to differentiate molecular characteristics between progenitor domains (*Tucker et al., 2008*; *Zechel et al., 2014*). More recent studies performed comprehensive scRNAseq experiments on the developing neocortex, but they did not focus on the ventral telencephalon or interneuron development (*Di Bella et al., 2021*; *Loo et al., 2019*; *Moreau et al., 2021*; *Telley et al., 2019*; *Telley et al., 2016*). scRNAseq studies that targeted GEs found that initial signatures of mature interneuron (IN) subtypes appear in postmitotic precursors, whereas very little transcriptional diversity was detected in VZ and SVZ progenitors (*Chen et al., 2017*; *Mayer et al., 2018*; *Mi et al., 2018*). However, these studies were underpowered for detecting potential transcriptional diversity in VZ neural progenitors because the GEs are comprised primarily of SVZ and mantle zone (MZ) cells at these mid-embryonic ages. In other central nervous system regions, transcriptionally heterogeneous VZ cell populations have been reported (*Johnson et al., 2015*; *Li et al., 2020*; *Ogawa et al., 2005*; *Yuzwa et al., 2017*). Thus, we still lack a definitive characterization of differential gene expression in VZ and SVZ neural progenitors in the embryonic mouse forebrain.

In this study, we performed scRNAseq analyses on the LGE, MGE, CGE, and cortex from E12.5 and E14.5 mice to identify spatial and temporal genetic heterogeneity of VZ and SVZ neural progenitors. To specifically increase the number of VZ cells, we utilized a reporter mouse line containing a destabilized VenusGFP protein that is driven by the *Nestin* promoter, which limits GFP leakiness into non-VZ cells and ensures GFP is a reliable readout of VZ cells (*Sunabori et al., 2008*). Using this mouse line, we observed significant transcriptional diversity of VZ cells within the ventral forebrain, both between different GEs as well as spatially segregated expression profiles of VZ cells within specific GEs. Many of these intriguing gene expression patterns were confirmed via multiplex in situ hybridizations. We also uncovered spatially-restricted gene expression patterns in the SVZ between and within the LGE, MGE, and CGE. Last, integrated transcriptional analysis of GE cells from E12.5 and E14.5 mice highlighted differential gene expression between VZ cells during these embryonic timepoints, which could have important implications for the well-described temporal changes in interneuron cell fate (*Bandler et al., 2017*; *Inan et al., 2012*; *Miyoshi and Fishell, 2011*; *Rymar and Sadikot, 2007*). Thus, our data reveal both spatial and temporal transcriptional diversity in distinct VZ and SVZ progenitor cell populations within the ganglionic eminences, which will further our understanding of neurogenesis during initial fate decisions in the ventral forebrain.

## Results

### Identification of distinct cell groups in the E12.5 mouse telencephalon

To characterize cellular heterogeneity in the developing telencephalon, we used the 10X Genomics scRNAseq platform to profile the transcriptome of cells from the LGE, MGE, CGE and CTX of E12.5 wild-type (WT) mice. At E12.5, neurogenesis has been occurring in the GEs for several days, with most cells residing in the SVZ and MZ with significantly fewer cycling VZ neural progenitors (*Turrero García and Harwell, 2017*; *Wichterle et al., 2001*). Conversely, the vast majority of E12.5 dorsal cortical cells are VZ neural progenitors, with a small number of layer VI projection neurons starting to emerge at this time (*Di Bella et al., 2021*; *Kwan et al., 2012*). To ensure that we collected a sufficient number of VZ cells from the GEs to identify transcriptional diversity in this population, we also dissected E12.5 brains from transgenic reporter mice that express destabilized Venus driven by the *Nestin* promoter and intronic enhancer (Nes-dVenus) (*Sunabori et al., 2008*; *Figure 1A–B*). We used fluorescence-activated cell sorting (FACS) to isolate GFP-positive cells from Nes-dVenus E12.5 mouse telencephalon (*Figure 1—figure supplement 1*). Note that ~ 87% of cortical cells are GFP-positive at E12.5 compared to 41–53% of GE cells, consistent with different proportions of VZ cells in these regions (*Figure 1—figure supplement 1B*). After filtering out non-viable outliers via the 10 X Genomics Cell Ranger pipeline, we obtained >84,000 potential cells (*Figure 1—figure supplement 2*). We then removed predicted doublets using *Doubletfinder* (*McGinnis et al., 2019*) and applied a stringent cutoff of >1500 genes per cell, which resulted in 36,428 E12.5 cells passing quality control that were used for downstream analysis. Cells harvested from both WT and Nes-dVenus mice were used for all subsequent analysis.

We used the Seurat software package (*Stuart et al., 2019*) to integrate these cells and identified 20 cell clusters that were largely segregated by brain regions, with clean separation of the CTX and MGE cells while there was greater overlap between the LGE and CGE populations (*Figure 1C*). We observed common VZ markers (*Nestin* and *Hes5*) and SVZ/early neurogenic markers (*Dcx* and *Ccnd2*) present in cells from all four regions, as well as GABAergic-specific (*Ascl1*, *Dlx5*, and *Gad2*) and glutamatergic-specific (*Pax6*, *Neurod6*, *Eomes*, *Tbr1*, and *Slc17a7*) genes restricted to their expected populations (*Figure 1D*). By comparing each region individually within the dataset, we observed many genes strongly enriched in the CTX, LGE, MGE, and CGE. While some of these genes have well-described roles in specific brain regions, we did uncover many genes whose regional restricted patterns have not been previously described, such as the alpha-internexin encoding gene *Ina* and Insulin like Growth Factor Binding Protein five encoding gene *Igfbp5* in the CGE (*Figure 1—figure supplement 3f*). Together, these data reveal different cohorts of telencephalic cells display distinct region-specific gene expression profiles exist at E12.5.

### Common neurogenic zones within each ganglionic eminence

To identify specific cell clusters in the GEs, we removed cortex-derived cells and identified 17 GE-derived cell clusters (*Figure 2A*). Based on *Nestin* and *Dcx* expression patterns, we identify seven clusters that likely represent VZ cells (high *Nestin*), six clusters that likely represent postmitotic cells (high *Dcx*), and four clusters that likely represent SVZ basal progenitors (moderate levels of both *Nestin* and *Dcx*). To confirm successful enrichment of *Nestin*-expressing VZ cells with the Nes-dVenus mouse, we plotted GE cells based on which mouse line they were derived from. This approach clearly demonstrates that the vast majority of high *Nestin*-expressing cells are indeed derived from the Nes-dVenus mouse, with significantly fewer VZ cells captured in the WT mice (*Figure 2A*). Violin plots revealed that most cells expressing VZ-enriched genes such as *Nestin*, *Hes5,* and *Ccnd1* were harvested from the Nes-dVenus mouse whereas a greater percentage of cells expressing more mature markers such as *Dcx*, *Tubb3,* and *Gad2* arose from the WT mice (*Figure 2—figure supplement 1A*). Thus, we have a significantly greater percentage of VZ cells compared to previous studies (*Mayer et al., 2018*; *Mi et al., 2018*; *Figure 2—figure supplement 2*), which provides us greater power to identify transcriptional heterogeneity and smaller subgroups in this VZ cell population that may have been previously overlooked.

The LGE, MGE, and CGE cells were individually extracted out and reanalyzed to reveal three common cell types shared between regions: (1) *Nestin*, *Hes5,* and *Olig2* expressing mitotic VZ cells, (2) *Ascl1*, *Ccnd2*, and *Gadd45g* expressing SVZ/BP cells, and (3) *Dcx*, *Sp9*, and *Mapt* expressing postmitotic SVZ and MZ cells (*Figure 2B* and *Figure 2—figure supplement 3*). Focusing on MGE-specific

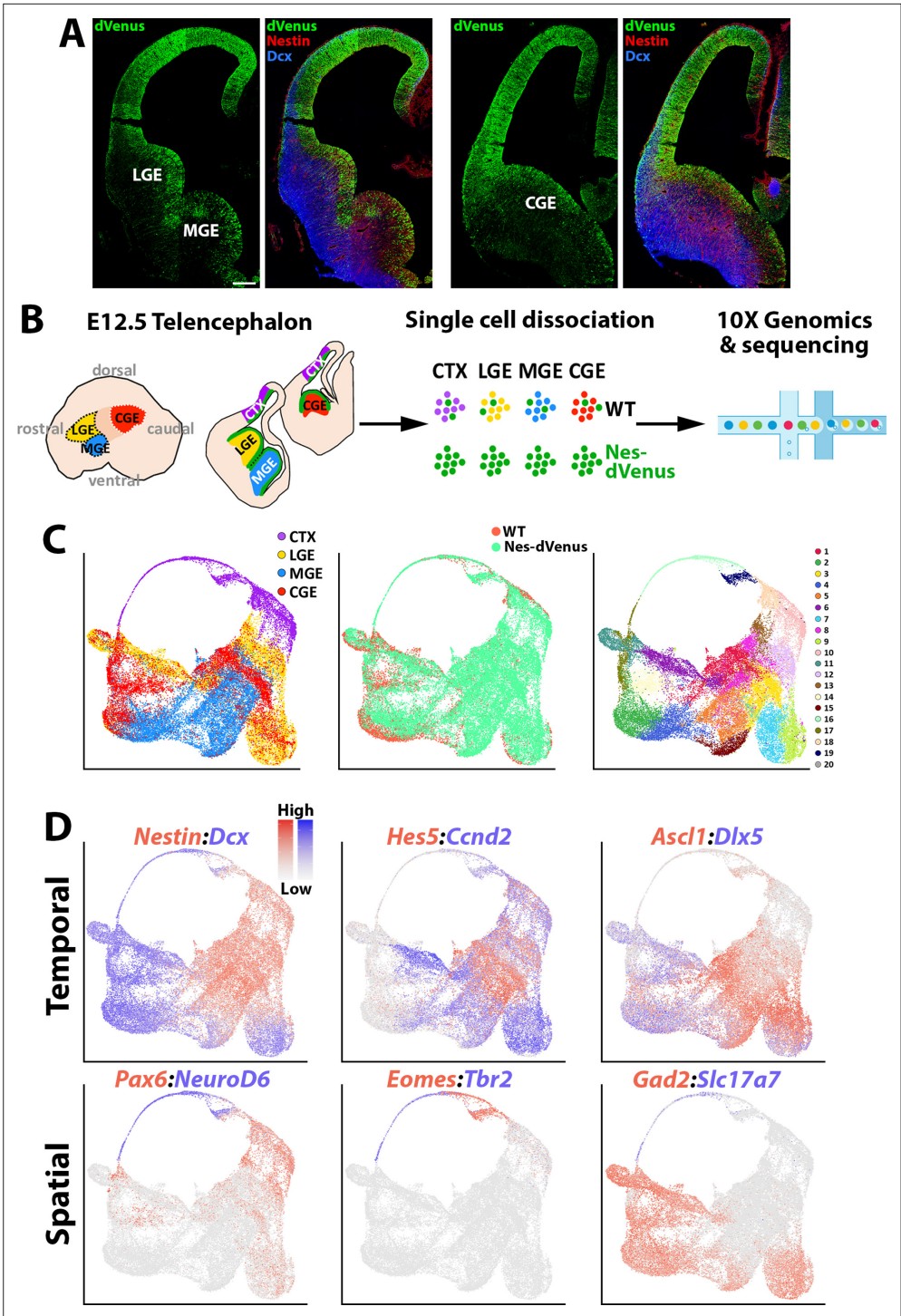

**Figure 1.** scRNAseq of distinct E12.5 forebrain regions from WT and Nes-dVenus mice. (**A**) Coronal section through the LGE/MGE (left) and CGE (right) of a E12.5 Nes-dVenus telencephalon immunostained for GFP (green), Nestin (Red) and Dcx (Blue). Scale bar = 200 μm. (**B**) Schematic of telencephalic cell dissection and single-cell dissociation. (**C**) Uniform manifold approximation and projection (UMAP) plots of all cells labeled by brain region (left), mouse line (middle) or putative cell clusters (right). (**D**) Representative genes displaying enriched temporal and/or spatial expression patterns. CTX, cortex; LGE, lateral ganglionic eminence; MGE, medial ganglionic eminence; CGE, caudal ganglionic eminence; WT, wild-type.

The online version of this article includes the following figure supplement(s) for figure 1:

**Figure supplement 1.** Flow cytometry plots depicting isolation of GFP-positive cells in Nes-dVenus mice.

*Figure 1 continued on next page*

*Figure 1 continued*

**Figure supplement 2.** Quality-control of sequenced cells.

**Figure supplement 3.** Identification of region-specific genes from each telencephalic region at E12.5.

genes, *Nkx2.1* was expressed by most cells within the MGE whereas *Lhx6* was restricted to postmitotic MGE cells, a subset of which also expressed the mature MGE-derived subtype marker *Sst* (***Figure 1— figure supplement 2B*** and ***Figure 2—figure supplement 3***). Within LGE and CGE clusters, *Ebf1*-positive, *Mapt*-positive and *Sp9*-negative clusters likely identify GABAergic projection neurons (***Nery et al., 2002***; ***Figure 2—figure supplement 3***). Also, *Npy*- and *Sst*-positive clusters were captured

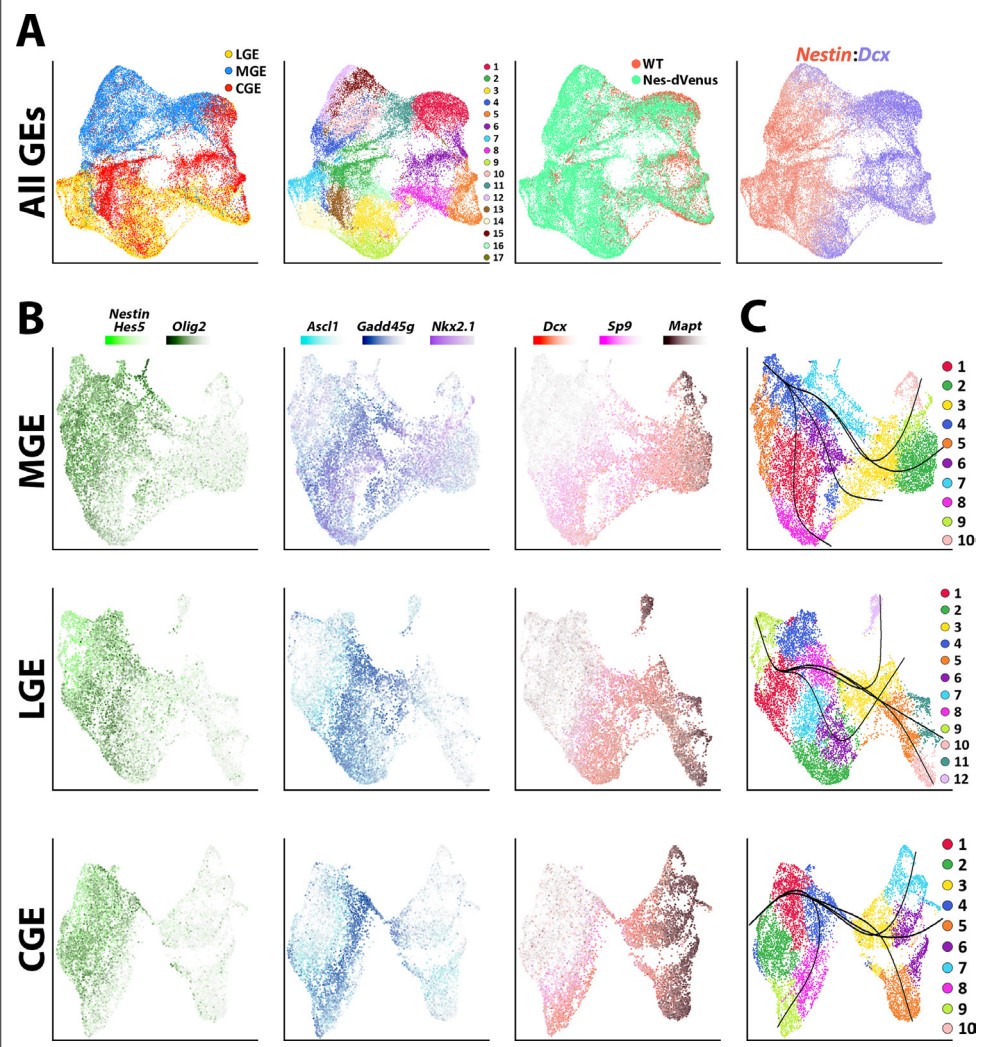

**Figure 2.** Common neurogenic lineages in each ganglionic eminence. (**A**) UMAP plots displaying all GE-derived cells annotated from left to right by: brain region, putative cell clusters, mouse strain and expression of VZ marker *Nestin* and SVZ/MZ marker *Dcx*. (**B**) UMAP visualizations show transcriptional diversity in each GE region. Several genes are depicted to highlight distinct neurogenic stages: *Nestin*, *Hes5*, and *Olig2* are indicative of VZ cells (green), *Asc1* and *Gadd45g* label intermediate SVZ cells (blue), and *Dcx*, *Sp9*, and *Mapt* depict postmitotic neuronal precursors (red). (**C**) UMAP plots of MGE, LGE, and CGE cells annotated via putative cell clusters and including Slingshot analyses depicting developmental progression through neurogenic stages.

The online version of this article includes the following figure supplement(s) for figure 2:

**Figure supplement 1.** Differential gene expression between mouse lines and brain regions.

**Figure supplement 2.** Integration and visualization of multiple embryonic mouse scRNAseq datasets.

**Figure supplement 3.** Genes defining common neurogenic cell types between brain regions.

within LGE and CGE dataset representing MGE-derived migrating cortical interneurons migrating through both regions towards the cortex (*Kessaris et al., 2014*; *Figure 2—figure supplement 3*). Pseudotime analysis with Slingshot (*Street et al., 2018*; *Trapnell et al., 2014*) showed clear trajectories in each region originating from *Nestin*-positive clusters progressing towards the postmitotic cell markers (*Figure 2C*). This analysis displays many branching patterns within VZ neural progenitors which could reflect distinct developmental trajectories among *Nestin*-positive VZ cells in GEs.

We next asked how GE-derived *Nestin*-expressing VZ neural progenitors are transcriptionally different from cycling *Dcx*-expressing SVZ cells and postmitotic SVZ/MZ cells. To specifically extract the VZ cells, we isolated cells that had a *Nestin* expression value above 1.5 from the log normalized GE count data (high *Nestin* cells). The value of this threshold was optimized by testing different expression values for subsetting, with the goal of harvesting a large number of high *Nestin*-expressing cells with minimal contamination of *Dcx*-expressing cells (*Figure 4—figure supplement 1A-B*). To obtain a relatively clean population of cycling SVZ/BP cells, we isolated cells that expressed *Dcx*, *Mki67* and *Ccnd2* but were negative for more mature cell markers like *Mapt* and *Rbfox3*. To enrich for postmitotic SVZ/MZ cells, we isolated cells that were negative for *Nestin*, *Hes5*, *Mki67*, *Ccnd1*, and *Ccnd2* but were positive for *Dcx* (*Figure 3A*). The subsetting value for postmitotic cells (high *Dcx* cells) was optimized with the similar approach to that of *Nestin*-expressing cell enrichment (*Figure 4—figure supplement 1C-D*). These three distinct cell populations retained their regional segregation and were grouped into 11 clusters (*Figure 3A*). Comparative analysis between the *Nestin*-positive, mitotic *Dcx*-positive, and postmitotic *Dcx*-positive groups showed clear transcriptional diversity, with 1,083 genes enriched in VZ cells, 330 genes upregulated in mitotic SVZ cells, and 1,320 genes upregulated in postmitotic SVZ/MZ cells (minimum log-fold change = 0.25) (*Supplementary file 1*). Highlighting the top 20 enriched genes for each cell type revealed several neural stem cell-associated genes found in VZ cells (*Fabp7*, *Hes5*, and *Ccnd1*), BP-associated genes for cycling SVZ cells (*Mpped2*, *Ccnd2*, and *Sp9*) and mature neuronal makers (*Mapt*, *Tubb2a,* and *Tubb3*) were detected in the postmitotic SVZ/MZ cells (*Figure 3B–C*). Each of the 11 identified cell clusters contained several enriched genes that correlated with their cell types, demonstrating GE-specific gene expression within each VZ, SVZ and MZ domain (*Figure 3D*). Our data uncovers transcriptional diversity within distinct neurogenic domains that may provide insight into how specific genes regulate early cell fate decisions within the embryo.

## Transcriptional heterogeneity throughout the ventricular zone

To uncover transcriptional heterogeneity specifically in VZ cells, we extracted and replotted the high *Nestin*-expressing cells (threshold >1.5) from the LGE, MGE, and CGE (*Figure 4*). This resulted in analysis of 9,308 high *Nestin*-expressing cells (3036 from LGE, 3890 from MGE, and 2302 from CGE). These cells were largely clustered based on brain region and were grouped into eight clusters. There was a large cohort of VZ-enriched genes that were restricted to one specific GE (*Figure 4B–C*), with some genes being confined to specific VZ clusters (subdomains) within a GE (*Figure 4D*). We selected ~20 genes that displayed intriguing spatial or cell-type-specific expression patterns within the VZ, the majority of which have not been previously described and visualized them with the RNAscope HiPlex in situ hybridization assay. We identified several pan-VZ genes such as *Ednrb*, *Pkdcc,* and *Nrarp* that are known to regulated by Wnt and Notch signaling pathways that are critical for embryonic development (*Krebs et al., 2001*; *Takeo et al., 2016*; *Vitorino et al., 2015*), as well as mitosis associated genes *Prc1*, *Cenpf,* and *Ube2c* (*Engeland, 2018*; *Figure 5—figure supplement 1A-C*). There were also genes that were strongly expressed in VZ cells in only two regions: *Ptx3* was expressed in VZ cells throughout the LGE and CGE yet absent from the MGE, while *Fgfr3* was strongly expressed in MGE and CGE VZ cells and absent in the LGE (*Figure 5A*, *Figure 2—figure supplement 1C* and *Figure 2—figure supplement 3C*). And several genes were restricted or strongly enriched in VZ cells in only one region, such as *Shisa2* and *Cntnap2* in the LGE, *Igfbp5* in the CGE, and *Asb4* in the MGE (*Figure 5B–D*, *Figure 2—figure supplement 1C* and *Figure 2—figure supplement 3C*).

We also observed many genes that had more refined spatially restricted expression patterns within various GEs, highlighting several examples below. *Gadd45g*, a gene that regulates stem cell proliferation (*Kaufmann and Niehrs, 2011*), displayed a salt-and-pepper profile with enriched expression in the dorsal LGE (dLGE) and ventral MGE and CGE (vMGE and vCGE) (*Figure 5—figure supplement 1C*). The UMAP plots of *Id4* and *Mest* look quite similar, but we observed slight differences in their VZ expression profiles. *Id4* was expressed in VZ cells throughout the LGE, CGE, and MGE, whereas

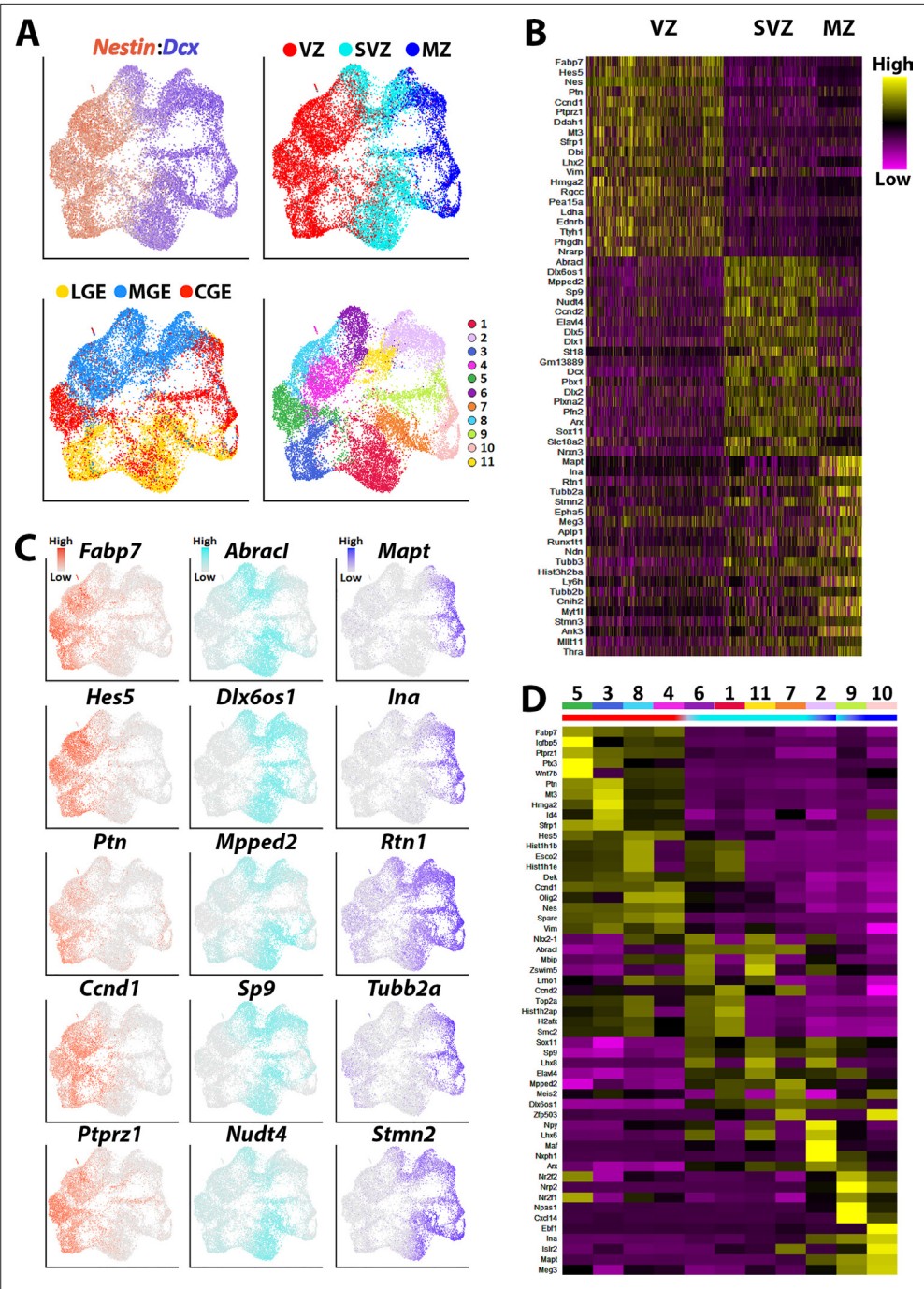

**Figure 3.** Transcriptional heterogeneity within the VZ, SVZ and MZ in E12.5 GEs. (**A**) High *Nestin*- and *Dcx*-expressing cells (> 1.5 fold expression above normalized dataset) were extracted from the GE populations and replotted. UMAP plots are annotated by: expression of *Nestin* and *Dcx* (upper left), brain region (lower left), separated into VZ (high-*Nestin*, red), SVZ (*Dcx*-, *Mki67*-, and *Ccnd2*-expressing, light blue) and MZ (*Dcx*-positive and *Ccnd2*- and *Mki67*-negative, dark blue) (upper right), and putative cell clusters (lower right). (**B**) Heatmap of genes enriched in VZ, SVZ and MZ cells. Each column represents expression in a single cell, color-coded as per the color scale. (**C**) UMAP plots depicting the top five differentially expressed genes (DEGs) in the VZ, SVZ, and MZ regions. (**D**) Heatmap showing expression of top 5 DEGs in each cell cluster from (**A**), with colored bar depicting whether each cluster contains predominantly VZ (red), SVZ (light blue), or MZ (dark blue) cells.

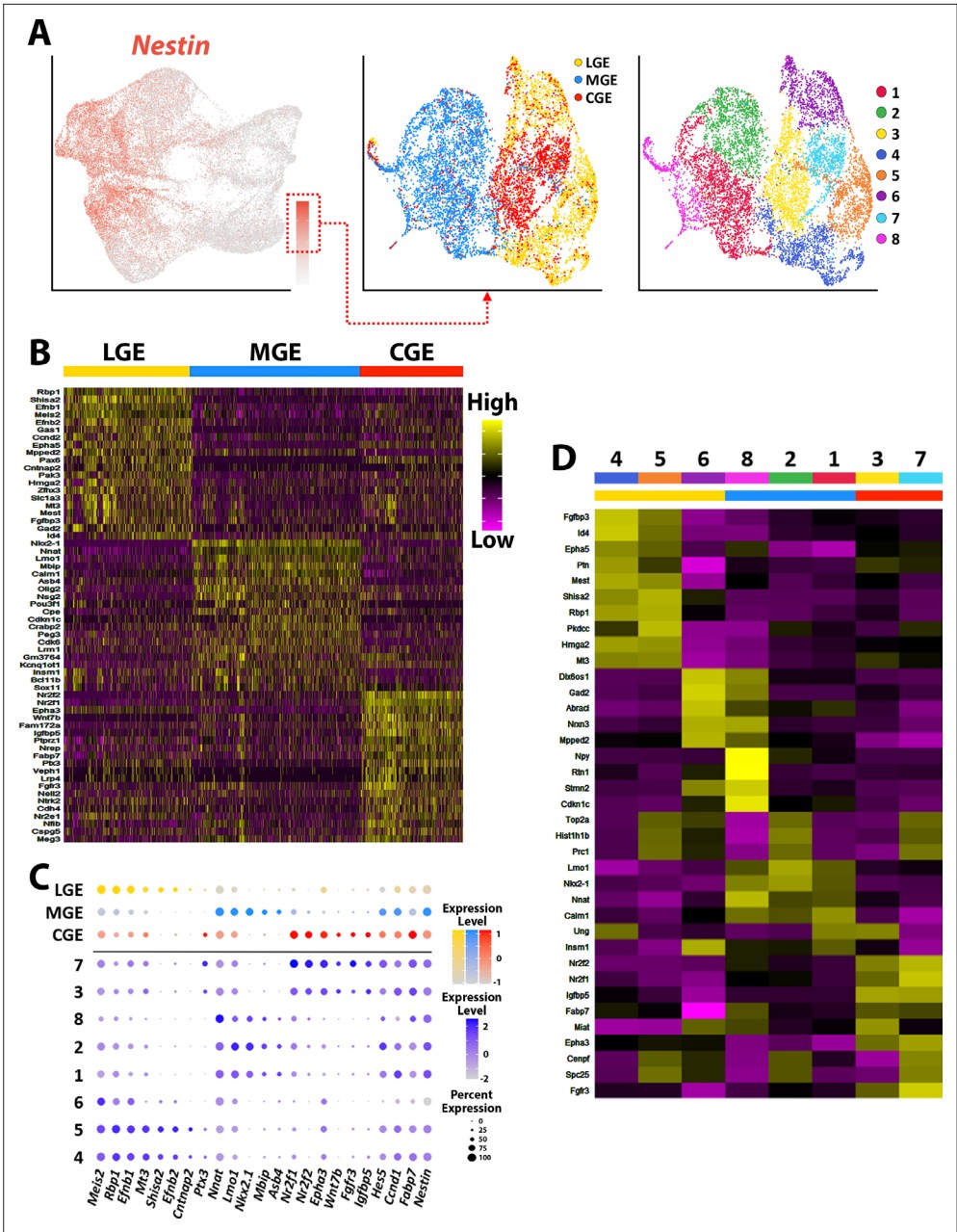

**Figure 4.** Transcriptional heterogeneity of *Nestin*-expressing VZ cells in the ganglionic eminences. (**A**) High *Nestin*-expressing (> 1.5 normalized dataset) cells were extracted from GE dataset and replotted as UMAP graphs annotated by brain region and putative cell clusters. (**B**) Heatmap of top 20 DEGs enriched in VZ cells from the LGE, MGE, and CGE. Each column represents expression in a single cell, color-coded as per the color scale. (**C**) Dot-plot depicting expression levels of DEGs within each brain region (LGE, MGE, and CGE) and cell cluster. (**D**) Heatmap showing expression of top 5 DEGs in each cluster from (**A**), with colored bar depicting whether each cluster contains cells from the LGE (yellow), MGE (blue), or CGE (red). Gray bar indicates cluster containing cells from all three GEs. Each column represents averaged expression in cells, color-coded as per the color scale.

The online version of this article includes the following figure supplement(s) for figure 4:

**Figure supplement 1.** Thresholding optimization for subsetting high *Nestin*- and *Dcx*-expressing GE cells.

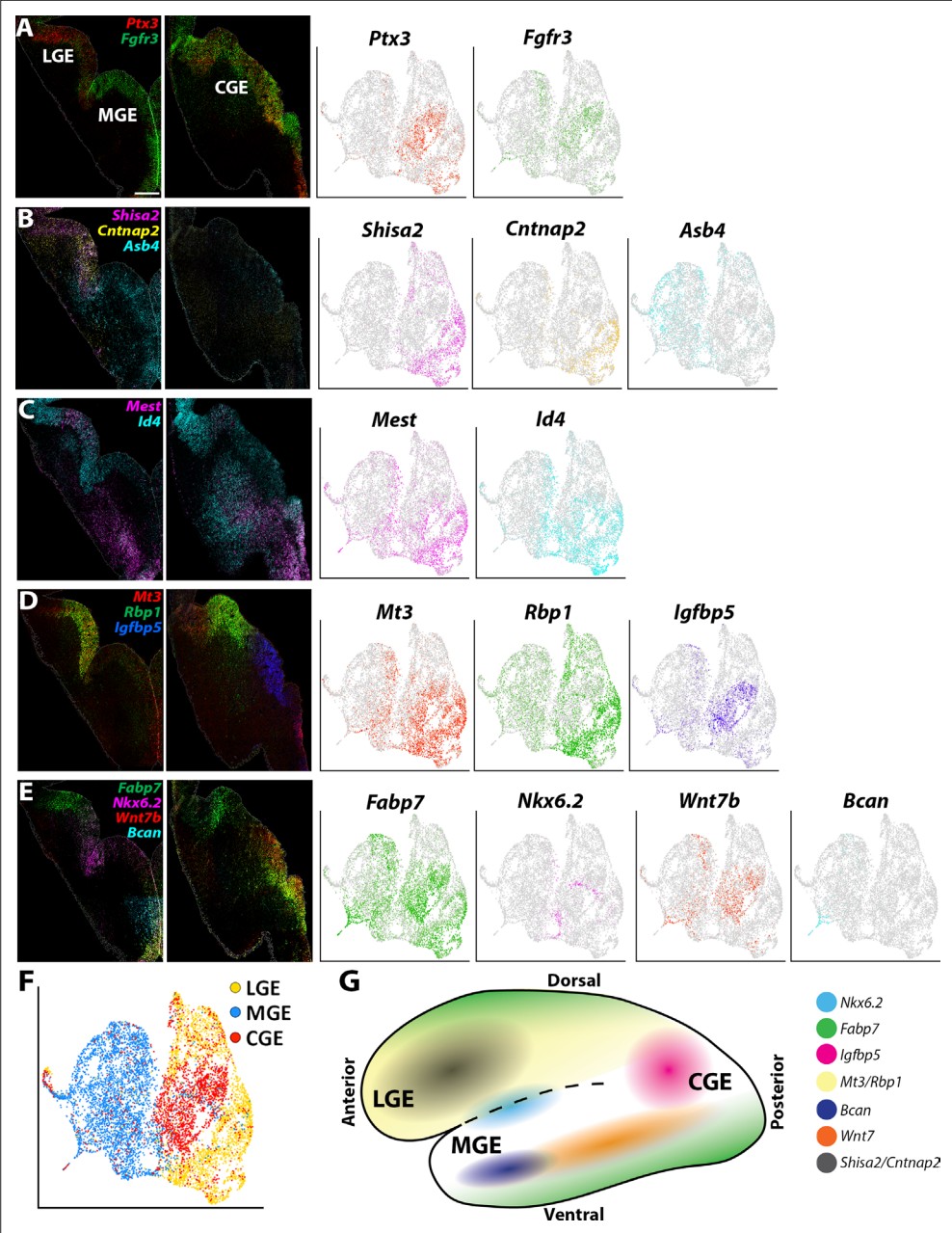

**Figure 5.** Expression profiles of spatially-enriched genes in VZ cells in the ganglionic eminences. (**A–E**) Left, RNAscope HiPlex and HiPlexUp in situ hybridization assays of differentially expressed VZ genes on E12.5 GE coronal sections. Right, the corresponding UMAP plots of VZ-enriched genes depicted in the in situs. Scale bar in A = 200 μm. (**F**) UMAP plot of *Nes*-enriched cells annotated by brain region. (**G**) Schematic of a whole mount view of the E12.5 ventral telencephalon depicting the approximate spatial location of VZ-enriched genes within the GEs.

The online version of this article includes the following figure supplement(s) for figure 5:

**Figure supplement 1.** Expression profiles of genes enriched in VZ cells throughout the GEs.

*Mest* was strongly enriched in the LGE and dorsal CGE (**Figure 5C**). These moderate differences highlight the need to confirm transcriptome expression profiles from scRNAseq experiments via in situ hybridizations.

*Mt3* and *Rbp1* are both strongly enriched in middle-to-ventral LGE VZ cells and in the dCGE, with only a smattering of *Mt3*- and *Rbp1*-expressing cells observed in the MGE (**Figure 5D**,

*Figure 2—figure supplement 1C* and *Figure 2—figure supplement 3C*). *Igfbp5* expression is exclusively restricted to vCGE VZ cells (*Figure 5D*, *Figure 2—figure supplement 1C* and *Figure 2—figure supplement 3C*). *Bcan* and *Wnt7b* are enriched in vMGE VZ cells, with *Wnt7b* expression extends into the vCGE, whereas *Nkx6.2* is expressed at the LGE/MGE boundary as previously described (*Sousa et al., 2009*; *Figure 5E*, *Figure 2—figure supplement 1C* and *Figure 2—figure supplement 3C*). Of note, *Fabp7* is strongly enriched in VZ cells in the dLGE, vMGE, and both dCGE and vCGE, as if forming longitudinal stripes along the dorsal and ventral GE boundaries (*Figure 5E*). In sum, our data reveal previously unidentified transcriptional heterogeneity within VZ cells throughout the GEs that define specific progenitor subdomains and likely regulate specific GABAergic cell types from these regions (*Figure 5G*).

## Transcriptional heterogeneity in SVZ/MZ cells within the ventral telencephalon

We utilized a similar approach to extract SVZ/MZ cells from the GEs that had high expression levels of *Dcx*, which displayed fairly clean segregation between LGE, MGE and CGE resulting in 11 cell clusters (total of 12,307 high *Dcx*-expressing cells with 3900 from LGE, 3902 from MGE and 4505 from CGE; *Figure 6—figure supplement 1A*). We observed clusters containing previously described region-enriched genes such *Nkx2.1*, *Lhx6* and *Lhx8* in the MGE, and *Nr2f1* and *Nr2f2* for CGE (*Kanatani et al., 2008*; *Lodato et al., 2011*); the expression profile of these genes was confirmed with in situ hybridizations (*Figure 6A–C*, *Figure 6—figure supplement 1A-C* and *Figure 6—figure supplement 2A*). CGE-enriched genes *Nr2f1* and *Nr2f2* are also expressed in the dLGE and POA, in agreement with previous observations (*Hu et al., 2017*; *Figure 6A*).

We also identified markers that separated cycling SVZ/BP cells from postmitotic MZ cells. Genes such as *Ascl1*, *Dlk2*, *E2f1*, and *Tcf4* are expressed in the VZ-SVZ boundary throughout all GEs and then downregulated in postmitotic MZ cells (*Figure 6—figure supplement 2B*). Conversely, *Zfhx3*, *Ina*, *Mapt*, *Mpped2*, and *Stmn2* are predominantly expressed in *Dcx*-positive postmitotic cells in the MZ layers, with *Zfhx3* and *Mpped2* expressed near the SVZ-MZ boundary, whereas *Ina*, *Mapt* and *Stmn2* are found in deeper MZ regions (*Figure 6B* and *Figure 6—figure supplement 2C*). These expression patterns reveal a conserved set of genes that are tightly regulated as cells transition from cycling progenitors to postmitotic GABAergic neurons throughout all GEs.

Other markers displayed more restricted expression patterns within the GEs. Most notably, we identified a 'stripe' of MZ cells in the ventral LGE that express a cohort of spatially enriched genes (*Mpped2*, *Pcsk1n*, *Zfp503*, *Zfhx3*, and *Aldh1a3*) (*Figure 6B–E* and *Figure 6—figure supplement 2C-D*). Both *Zfhx3* and *Zfp503/Nolz1* have recently been implicated in development of striatal medium spiny neurons (MSNs) (*Soleilhavoup et al., 2020*; *Zhang et al., 2019*), and *Aldh1a3/Raldh3* is critical for generation of LGE-derived neurons (*Chatzi et al., 2011*). *Ebf1*, a marker for direct pathway striatal neurons (*Lobo et al., 2008*), is also expressed in this region but not as tightly restricted to this narrow domain as the other genes (*Figure 6D*). Thus, this vLGE population likely represents future striatal MSNs, possibly D1-expressing MSNs (*Zhang et al., 2019*).

Focusing specifically on the CGE, *Cxcl14* is expressed by many CGE-derived interneurons in the mature cortex and hippocampus (*Harris et al., 2018*; *Tasic et al., 2018*), although it was reported to not be expressed in the embryonic mouse brain (*Wei et al., 2019*). We detected *Cxcl14* expression in the MZ of the CGE with a bias toward the vCGE (*Figure 6D*), a similar expression profile to *Zfhx3*. Conversely, the non-coding RNA *Rmst* is enriched in the dCGE and displays a complementary pattern to *Nr2f1/2* (*Figure 6A*). Several other genes are also enriched in the SVZ/MZ region of the dCGE, such as *Ccnd2* and *Flrt2* (*Figure 6C–F*). Thus, similar to the MGE, there are differential expression patterns along the dorsal-ventral axis of the CGE. How these distinct VZ and SVZ/MZ subdomains related to CGE-derived interneuron fate remains unknown.

To highlight a few more notable expression patterns, the pro-neural gene *Pou3f1/Oct-6* is strongly enriched in LGE SVZ adjacent to the *Ebf1* domain and is not expressed in the MGE or CGE (*Figure 6D*). *Nrp2* is enriched in deep MZ cells within the dLGE, dMGE, and vCGE, whereas *Pcsk1n* displays a complementary pattern in the vLGE and vMGE but is not expressed within the CGE (*Figure 6—figure supplement 2D*). *Ccnd2* and *Flrt2* were enriched in the SVZ layer of both LGE and CGE with significantly weaker expression in the MGE (*Figure 6C–E*). Curiously, *Rprm* is enriched in the SVZ of both the dMGE and a small group of cells in the most ventral CGE (*Figure 6C*). In sum, these single-cell

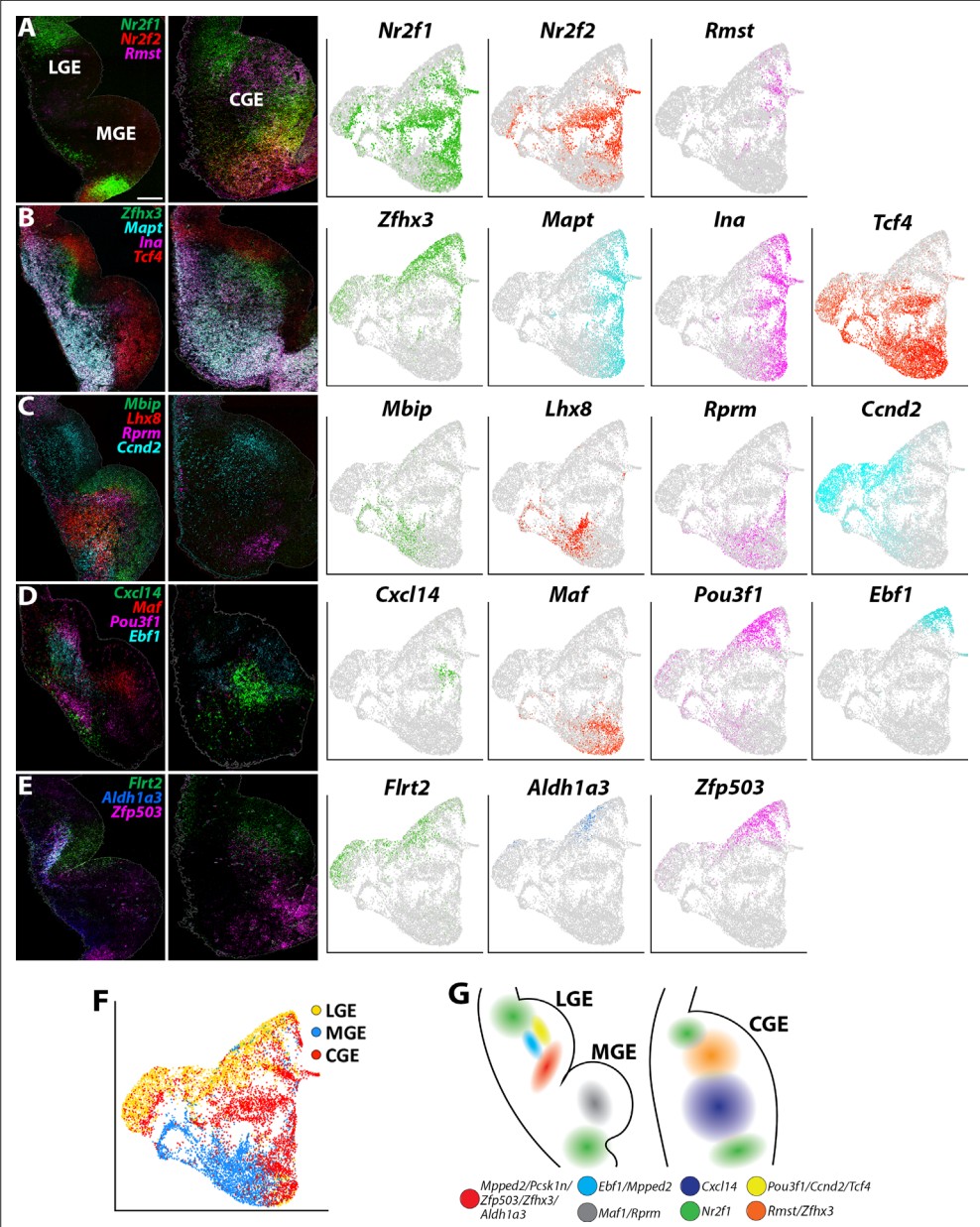

**Figure 6.** Expression profiles of spatially-enriched genes in SVZ/MZ cells in the ganglionic eminences. (**A–E**) Left, RNAscope HiPlex and HiPlexUp in situ hybridization assays of differentially expressed SVZ/MZ genes on E12.5 GE coronal sections. Right, the corresponding UMAP plots of SVZ/MZ-enriched genes depicted in the in situs. Scale bar in A = 200 μm. (**F**) UMAP plot of *Dcx*-enriched cells annotated by brain region. (**G**) Schematic of a coronal section through the E12.5 ventral telencephalon depicting the approximate spatial location of SVZ/MZ-enriched genes within the GEs.

The online version of this article includes the following figure supplement(s) for figure 6:

**Figure supplement 1.** Characterization of genes enriched in high *Dcx*-expressing cells.

**Figure supplement 2.** Expression profiles of additional SVZ/MZ-enriched genes in the ganglionic eminences.

gene expression profiles provide new insight into the transcriptional diversity in *Dcx*-positive SVZ/MZ postmitotic GE cells (*Figure 6G*) and revealed subdomains of genetically defined SVZ/MZ cells that have not been previously described.

## Developmental transition of ventral telencephalic VZ neural progenitors over time

There is ample evidence that the capacity for neural progenitors to generate specific neuronal subtypes changes over time (*Gal et al., 2006*; *Pilz et al., 2013*). In the MGE, production of SST-expressing interneurons is significantly decreased at E14.5 compared to E12.5 (*Bandler et al., 2017*; *Inan et al., 2012*; *Miyoshi and Fishell, 2011*). To characterize transcriptional changes in GE progenitors over time, we collected LGE, MGE and CGE cells from E14.5 WT and Nes-dVenus mice. These cells were largely segregated by brain region and consisted of 19 cell clusters (*Figure 7—figure supplement 1A-C*). As with the E12.5 population, significant heterogeneity was still observed when extracting out the highest *Nestin*- and *Dcx*-expressing cells (*Figure 7—figure supplement 1D-G*).

To compare the E12.5 and E14.5 cells, we integrated these datasets together. The majority of postmitotic cell clusters consisted of E14.5 cells, and most *Nestin*-expressing cells were derived from the Nes-dVenus mouse (*Figure 7A*). These integrated populations were still primarily segregated based on region and were divided into seventeen different clusters (*Figure 7B–C*). We isolated strong *Nestin*- and *Dcx*-expressing cells from the integrated dataset and performed a differential expression analysis on both sets of cells (*Figure 7D–F*). High *Dcx*-expressing from E12.5 and E14.5 were nearly completely overlapping with no significant differences, indicating that the global population of postmitotic cells from GEs are transcriptionally very similar at these ages (*Figure 7F*). However, when we compared the high *Nestin*-expressing cells, we observed more age-specific segregation in the dot plot clusters, with one subdomain predominantly consisting of E12.5 cells and another population containing E14.5 cells (*Figure 7E*).

To compare these two cell clusters, we removed the E12.5 cells from the E14.5 cloud (and vice versa) and identified many genes enriched specifically in E12.5 or E14.5 *Nestin*-expressing cells (*Figure 7G*). Changes in VZ gene expression between these two timepoints were also observed within specific GEs (*Figure 7H*). Many genes enriched in E14.5 high *Nestin*-expressing are commonly associated with BPs or postmitotic cells such as *Dcx*, *Tubb3*, *Mpped2*, *Arx,* and *Nrxn3* (*Figure 7G*). The adult neural progenitor marker *Igfbpl1* (*Artegiani et al., 2017*) was also upregulated in E14.5 VZ cells when compared to E12.5 (*Figure 7G*). To ensure that the dVenus is not 'leaking' into non-VZ cells at E14.5 and contaminating this dataset with SVZ/BP cells, we confirmed that many previously reported pan-VZ cell markers are still expressed in the E14.5 high *Nestin*-expressing cells (*Figure 7—figure supplement 2*). Taken together, this data suggests that while *Dcx*-positive cells are transcriptionally very similar between E12.5 and E14.5, *Nestin*-positive VZ cells in the GEs display a transition in their gene expression profiles, with a greater number of neurogenic and postmitotic genes present in E14.5 VZ cells.

We next validated some of the differentially expressed genes between E12.5 and E14.5 via RNAscope HiPlexUp. As predicted from the scRNAseq data (*Figure 7G*), *Sfrp1* and *Sparc* were strongly downregulated in E14.5 VZ GE cells compared to E12.5 (*Figure 8A*). *Mir124a-1hg* was strongly enriched in E14.5 VZ cells in the LGE compared to E12.5, agreeing the predictions from the scRNAseq data (*Figure 7G*), whereas *Gucy1a1* expression was upregulated in the SVZ of E14.5 LGE and CGE (*Figure 8B*). *Id4* was strongly downregulated in E14.5 VZ cells in the LGE and MGE, but was significantly upregulated in E14.5 SVZ cells in the LGE and CGE (*Figure 8C*). Notably, several of these genes displayed weaker expression in the MGE SVZ at E14.5 compared to E12.5 (*Mir124a-1hg*, *Gucy1a1* and *Cited2*; *Figure 8B–C*). Focusing specifically on the CGE, we observe a strong downregulation of *Hmga2* and *Igfbp5* in E14.5 VZ cells and an upregulation of *Sox6* in both VZ and SVZ cells (*Figure 8D*). We also observed numerous gene expression patterns that were consistent between E12.5 and E14.5, such as *Slc1a3* in the dCGE VZ, *Sp8* in the dLGE SVZ and *Ebf1* throughout the LGE SVZ/MZ (*Figure 8—figure supplement 1*). These in situ hybridizations confirm our scRNAseq data indicating dynamic changes in gene expression patterns in VZ and SVZ cells between E12.5 and E14.5, likely reflecting changes in cell fate trajectories throughout embryonic development.

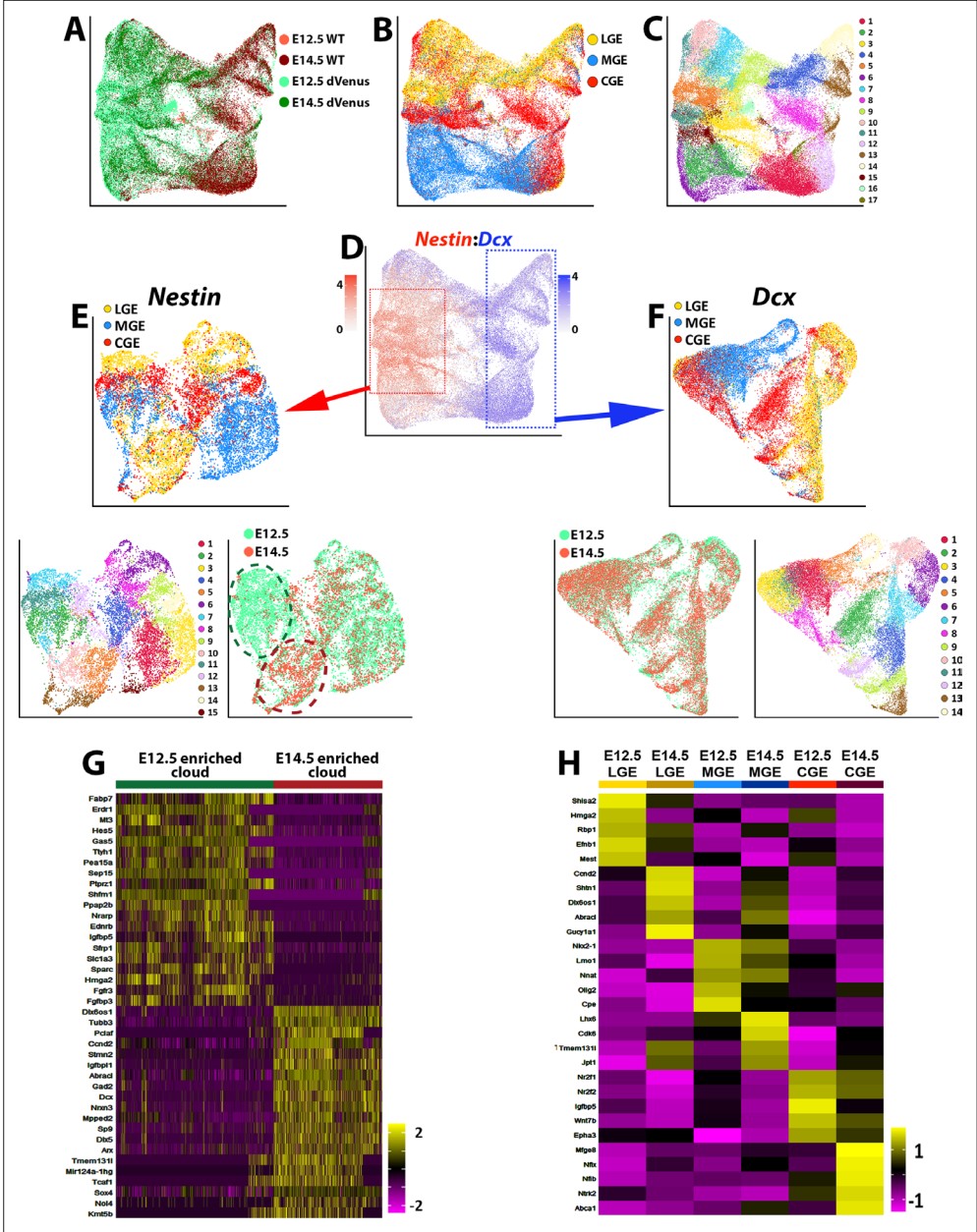

**Figure 7.** Comparison of transcriptional changes between E12.5 and E14.5 ventral telencephalon. (**A–C**) UMAP plots of all E12.5 and E14.5 cells annotated by mouse line and embryonic timepoint (**A**), brain region (**B**), and putative cell clusters (**C**). (**D**) UMAP plot of E12.5 and E14.5 cells depicting expression levels of *Nestin* and *Dcx*. Red box = approximate location of high *Nestin*-expressing cells (> 1.5 normalized dataset), blue box = approximate location of high *Dcx*-expressing cells (> 1.5 normalized dataset). (**E**) UMAP plot depicting high *Nestin*-expressing cells annotated by brain region (top), putative cell clusters (lower left) and timepoint (lower right). Note the clouds that are strongly enriched for E12.5 cells (dark green oval) and E14.5 cells (dark red oval). (**F**) UMAP plot depicting high *Dcx*-expressing cells annotated by brain region (top), putative cell clusters (lower right) and timepoint (lower left). Note that E12.5 and E14.5 cells are largely overlapping populations with no clear differential clustering. (**G**) Heatmap depicting the top 20 DEGs between the E12.5-enriched (dark green oval) and E14.5-enriched (dark red oval) clouds of VZ cells from (**E**). The E14.5-derived cells were removed from the E12.5-enriched cloud, and vice versa. Each column represents expression in a single cell, color-coded as per the color scale. (**H**) Heatmap depicting the top 5 DEGs in VZ cells from each GE region at E12.5 and E14.5. Each column represents averaged expression in cells, color-coded as per the color scale.

The online version of this article includes the following figure supplement(s) for figure 7:

*Figure 7 continued on next page*

**Figure supplement 1.** Single-cell transcriptional profiling of E14.5 ganglionic eminences.

**Figure supplement 2.** Expression of pan-VZ genes in high *Nestin*-expressing E12.5 and E14.5 cells.

## Discussion

We performed scRNAseq on distinct neurogenic forebrain regions to generate a comprehensive transcriptional overview of embryonic telencephalic progenitors. By utilizing the Nes-dVenus mouse, we were able to significantly increase the number of VZ cells and differentiate between cycling VZ cells (RGCs/APs) and cycling SVZ cells (BPs), a distinction that had not been explored in previous forebrain scRNAseq studies. Validating scRNA-seq findings via in situ hybridizations is critical to ensure accuracy of the dataset and to characterize additional spatial heterogeneity that is not obtainable from scRNA-seq data alone. Our in situs revealed a rich diversity of gene expression profiles in progenitors throughout the VZ and SVZ/MZ of the ventral forebrain. These insights could increase our understanding of GABAergic neuronal cell type patterning during neurogenesis and have important implications for understanding neurodevelopmental disorders because many disease-associated genes are enriched in radial glia/VZ cells and interneuron progenitors (*Schork et al., 2019*; *Trevino et al., 2020*).

We uncovered striking spatial transcriptional diversity throughout the VZ layers, with many genes displaying spatially-restricted expression patterns that had not been previously reported. We observe a variety of expression patterns, with some genes displaying sharp boundaries within subdomains of GEs while other genes are expressed as gradients across GEs. Some genes were restricted to one or two GEs, with several genes restricted or strongly enriched in the CGE. There are minimal tools to genetically target CGE-specific cell types during embryogenesis because of the lack of established

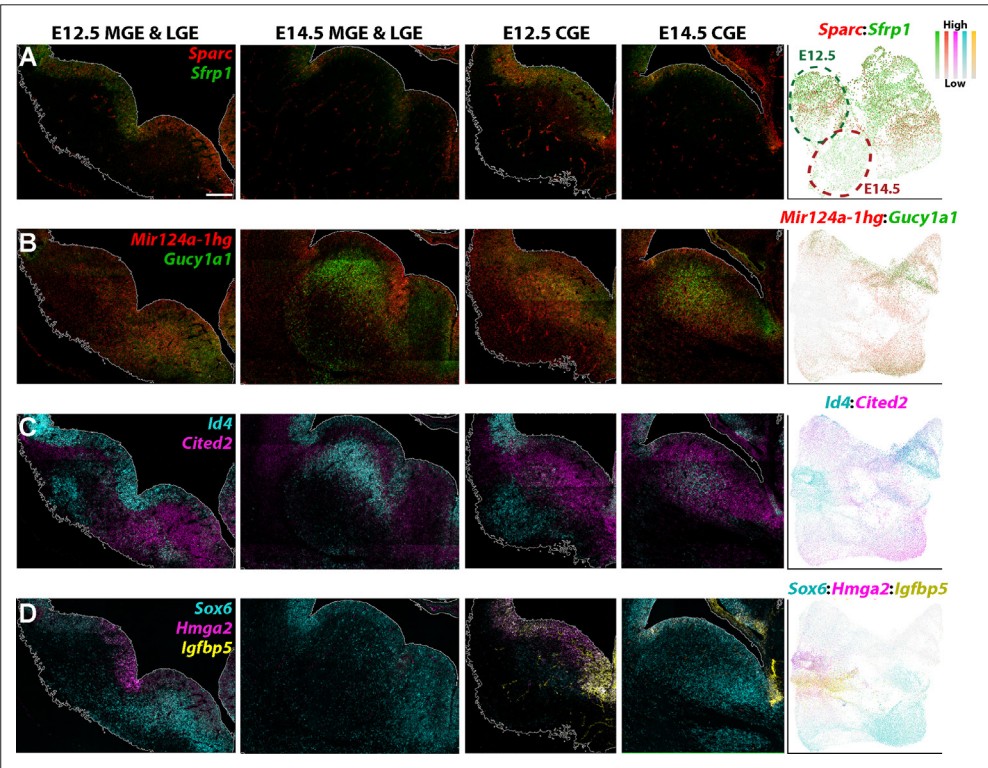

**Figure 8.** Genes with differential expression profiles between E12.5 and E14.5 ganglionic eminences. (**A–D**) RNAscope HiPlexUp assay of genes differentially expressed in VZ and SVZ cells between E12.5 and E14.5 on coronal sections through the GEs. Dashed circles on the UMAP plot (**A**) indicate the clouds that are strongly enriched for E12.5 cells (dark green) and E14.5 cells (dark red). Scale bar in A = 200 µm. Dotplot in (**A**) contains high *Nestin*-expressing cells from E12.5 and E14.5, dotplots in B-D consist of all E12.5 and E14.5 cells.

The online version of this article includes the following figure supplement(s) for figure 8:

**Figure supplement 1.** Genes with similar expression profiles between E12.5 and E14.5 ganglionic eminences.

CGE-specific genes, and even some traditionally described CGE-specific genes such as *Nr2f1* and *Nr2f2* are also expressed in the posterior MGE (*Hu et al., 2017*). Our finding that Igfbp5 is restricted to the CGE at E12.5 (*Figure 5D*) warrants further investigation into Igfbp5 as a promising tool to label CGE-derived cells.

Numerous genes display restricted expression patterns that are enriched in dorsal or ventral VZ cells in specific structures. This is particularly relevant due to the ample evidence that certain spatial domains are biased to generate particular interneuron subtypes (*Brandão and Romcy-Pereira, 2015*; *Flames et al., 2007*; *McKinsey et al., 2013*; *Tyson et al., 2015*; *Wonders et al., 2008*; *Xu et al., 2010*). While a relationship between spatial domain and interneuron subtype has been described in the MGE, whether distinct CGE-derived interneuron subtypes arise from specific CGE regions remains unknown. Several genes such as *Rbp1*, *Id4,* and *Wnt7b* are enriched in subdomains of VZ cells within the CGE, which may indicate that CGE subdomains preferentially give rise to specific interneuron subtypes. Another intriguing expression profile was *Fabp7*, which is traditionally believed to be a pan-VZ/radial glia cell marker (*Arai et al., 2005*; *Schmid et al., 2006*; *Yun et al., 2012*; *Yuzwa et al., 2017*). However, in the ventral telencephalon, *Fabp7* is restricted to the dorsal and ventral GE boundaries, as if demarcating the GEs from the surrounding forebrain.

We also detected transcriptionally defined subdomains within the SVZ/MZ layers to a greater extent than was previously appreciated. For example, *Mbip* and *Rprm* show a complimentary expression profile at the VZ/SVZ and SVZ/MZ border in MGE, respectively. We observed several genes (*Mpped2*, *Pcsk1n*, *Aldh1a3*, *Zfhx3,* and *Zfp503*) strongly enriched at the SVZ/MZ boundary in the ventral LGE (*Figure 6* and *Figure 6—figure supplement 2*), which likely represent future medium spiny neurons of the striatum (*Chatzi et al., 2011*; *Soleilhavoup et al., 2020*; *Zhang et al., 2019*). Within the CGE, *Nrp2, Rprm,* and *Zfp503* are enriched in the ventral region whereas *Mpped2*, *Ccnd2,* and *Flrt2* are enriched in the SVZ/MZ of the dorsal CGE. As cell fates become more defined as cells transition from cycling to postmitotic at this SVZ/MZ border, further work is needed to characterize the role of these spatially restricted genes in this process.

Additionally, we also identify a transition in the transcriptional profile of VZ cells from E12.5 to E14.5 that was not observed in SVZ/MZ cells, which were transcriptionally very similar between the two ages. Temporal transcriptional changes in VZ cells were most prominent in the LGE according to our comparative and RNAscope analyses (*Figures 7E and 8*). In general, E14.5 VZ cells from the GEs begin to express genes that are indicative of more mature cells, such as *Dcx, Arx,* and *Gad2*. Many of the VZ cell-enriched genes at E14.5 are also strongly expressed in the SVZ layer (*Figures 7G–H , and 8B*). Perhaps environment-dependent signals arising at later embryonic timepoints drive E14.5 VZ cells to become 'extraverted' when compared to their 'introverted' E12.5 counterparts (*Telley et al., 2019*). These transcriptional changes are occurring during a simultaneous increase in the number of SVZ cells/BPs with a proportional decrease in the number of proliferative VZ cells, as most MGE VZ cells give rise to SVZ/BP cells by E14.5 (*Glickstein et al., 2007*; *Petros et al., 2015*; *Tsoa et al., 2014*). Gene expression changes in VZ cells over time could guide this transition in cell cycle dynamics and in part regulate temporal changes in cell fate throughout neurogenesis (*Bandler et al., 2017*; *Inan et al., 2012*; *Miyoshi and Fishell, 2011*). It will be intriguing to explore how these changes in gene expression in VZ cells over time alters their neurogenic potential and cell fate capacity.

In sum, this study characterized the gene expression profiles of VZ and SVZ cells in distinct neurogenic regions of the embryonic telencephalon, providing important insights into the transcriptome of distinct regional domains that may guide early neuronal fate decisions. There is growing evidence that transcriptional diversity in distinct progenitor subtypes is critical for generating neuronal diversity (*Cheung et al., 2010*; *Molnár et al., 2006*). Hence, characterizing the developmental expression profiles in distinct progenitor cells in different brain regions during neurogenesis will further explain origins of cortical interneuron subtypes and how the neocortex is formed or altered in neurodevelopmental disorders.

## Materials and methods

**Key resources table**

| Reagent type (species) or resource | Designation | Source or reference | Identifiers | Additional information |
|---|---|---|---|---|
| Strain, strain background (*Mus musculus*) | Nes-dVenus | RIKEN BioResource Research Center | C57BL/6-Tg(Nes-d4YFP*)1HOkn, RRID:IMSR_RBRC04058 | Pooled sexes |
| Strain, strain background (*Mus musculus*) | Wild-Type | The Jackson Laboratory | C57BL/6 J 000664, RRID:IMSR_JAX:000664 | Pooled sexes |
| Sequence-based reagent | Chromium Next GEM Single Cell 3' Kit | 10X Genomics | v2 (120267) v3 (1000092) | v2 & v3 |
| Commercial assay or kit | RNAscope HiPlex Assays | Advanced Cell Diagnostics | 324,102 | v1 |
| Antibody | Anti-GFP (Rabbit polyclonal) | Invitrogen | A11122, RRID:AB_221569 | 1:400 |
| Antibody | Anti-Doublecortin (Chicken polyclonal) | Abcam | AB153668, RRID:AB_2728759 | 1:1,000 |
| Antibody | Anti-Nestin (Rat Monoclonal) | Millipore | MAB353, RRID:AB_94911 | 1:100 |
| Software, algorithm | Cell Ranger | 10X Genomics | RRID:SCR_017344 | v3.0.0 |
| Software, algorithm | Seurat | the Satija Lab at the New York Genome Center | RRID:SCR_007322 | v3.0.0 |
| Software, algorithm | Slingshot | Kelly Street and Sandrine Dudoit at UC Berkeley | RRID:SCR_017012 | https://github.com/kstreet13/slingshotv2.2.0 |
| Software, algorithm | DoubletFinder | Christopher S. McGinnis and Zev J. Gartner at UCSF | RRID:SCR_018771 | https://github.com/chris-mcginnis-ucsf/DoubletFinder v3.0 |
| Software, algorithm | RNAscope HiPlex Image Registration | Advanced Cell Diagnostics | | v1 and v2 |
| Software, algorithm | Photoshop CC | Adobe | RRID:SCR_014199 | 20.0.9 |
| Software, algorithm | ImageJ | National Health Institution | RRID:SCR_003070 | http://imagej.nih.gov/ij/ |

## Animals

All mouse colonies were maintained in accordance with protocols approved by the Animal Care and Use Committee at the *Eunice Kennedy Shriver* National Institute of Child Health and Human Development (NICHD). Wild-type (WT) C57BL/6 mice were obtained from The Jackson Laboratory whereas Nestin-d4-Venus (Nes-dVenus) mice were provided by RIKEN BioResource Research Center (*Sunabori et al., 2008*). For all embryonic experiments, the day on which a vaginal plug was found was considered as embryonic day (E) 0.5.

## Single-cell isolation

WT and Nes-dVenus pregnant dams at stage E12.5 and E14.5 were euthanized with Euthasol. Embryos were removed and incubated on ice in oxygenated artificial cerebrospinal fluid (ACSF) throughout dissection. ACSF, in mM: 87 NaCl, 26 NaHCO$_3$, 2.5 KCl, 1.25 NaH$_2$PO$_4$, 0.5 CaCl$_2$, 7 MgCl$_2$, 10 glucose, 75 sucrose saturated with 95% O$_2$, 5% CO$_2$, pH 7.4. Cortices and all three GEs were micro-dissected and tissue collected into labeled tubes with ACSF. Tissue was pooled from $\geq$ 4 embryos for each experiment and was then enzymatically dissociated with 1 mg/ml of Pronase (Sigma-Aldrich #10165921001) in ACSF for 15–20 min. Pronase solution was removed and 1–2 ml of reconstitution solution (ACSF +1:100 fetal bovine serum (FBS) +0.01% DNase) was added to each tube before mechanically dissociating with fire-polished glass pipettes of large, medium and small-bore openings. DAPI (1 μl) and DRAQ5 (5 μM, ThermoFisher #62251) were then added and cell solution was passed through a pre-wetted 35 μm filter prior to sorting (Nes-dVenus) or cell counting (WT).

## Fluorescence-activated cell sorting

Dissociated cell solution from Nes-dVenus mice were sorted with Beckman Coulter MoFlo Astrios cell sorter or Sony SH800S sorter with 100 μm chips. Cells were first gated with forward scatter (FSC) vs.

side scatter (SSC) to remove debris, then gated with DAPI vs. DRAQ5 to select for live cells (DRAQ5$^+$/ DAPI$^-$), then gated with GFP 488 vs. FSC to harvest GFP-expressing cells. Cells were collected into DNA LoBind microcentrifuge tubes (Eppendorf #022431021) containing cold oxygenated ACSF supplemented with 1% FBS.

## Single-cell RNA sequencing and library generation

For each experiment, ~ 15,000 cells from WT or sorted Nes-dVenus mice were run through the 10X Genomics Single Cell controller. Chromium Single Cell 3' GEM (versions 2 and 3), Library and Gel Bead Kits were used according to the manufacturer's instructions. Libraries were sequenced on Illumina HiSeq 2500 by the NICHD Molecular Genomics Core. For the E12.5 MGE/LGE/CGE, three biological replicates were used to make independent libraries (1 WT and 2 Nes-dVenus). For the E12.5 CTX and the E14.5 MGE/LGE/CGE samples, two replicates were used for library construction (1 WT and 1 Nes-dVenus). Library quality and DNA content were assessed using a BioAnalyzer (Agilent) and Qubit (ThermoFisher), respectively. Libraries from each experiment were balanced by DNA quantity, pooled, and sequenced on an Illumina HiSeq 2500 by the NICHD Molecular Genomics Core.

## Single-cell RNA sequencing analysis

Reads were aligned to the mouse genome (mm10) before generating a gene by barcode count matrix using Cell Ranger (10X Genomics) with default parameters. In total, ~120,000 cells were obtained from the original Cell Ranger pipeline. To analyze high quality cells only, we first removed cells that had unique molecular identifier (UMI) counts > 4500 or < 1500. We also ran *Doubletfinder* (**McGinnis et al., 2019**) to identify doublets and removed data points when the doublet score was greater than 0.3 from further analysis in each of the datasets. Cells were further filtered based on the percentage of UMI counts associated with mitochondrial mapped reeds and number of detected genes per cell, and we removed cells > 3 median absolute deviations (MADs) from the median of the total population. For cells derived from WT mice, an additional step was taken by filtering out cells based on the percentage of UMI associated with hemoglobin subunit beta (*Hbb*) transcripts (since Nes-dVenus cells were sorted, they did not require this additional filtering step). Filtration of cells based on quality control metrics, data normalization, scaling, dimensionality reduction, highly variable feature detections, population subsetting and data integration were all performed with the Seurat package using the standard gene expression workflow and default parameters except when scaling counts, where the mitochondrial transcript and ribosomal transcript percentages were regressed out, before non-linear dimensional reduction and community detection were performed (**Stuart et al., 2019**). Applying these stringent filtration steps resulted in a total of 36,428 E12.5 cells and 24,218 E14.5 cells that were analyzed in this manuscript, with the following breakdown: E12.5: 12,052 MGE cells, 10,316 LGE cells, 9275 CGE cells, 4785 CTX cells. E14.5: 8389 MGE cells, 8658 LGE cells, 7171 CGE cells.

To characterize lineage arising from progenitors, Slingshot was used for single cell trajectory inference using UMAP projections (**Street et al., 2018**). For the comparative analysis of VZ, SVZ, and MZ cells, the effects of cell cycle heterogeneity were mitigated by calculating cell-cycle phase scores based on known canonical markers (**Nestorowa et al., 2016**) and the scores were regressed out during the data scaling using ScaleData. To distinguish cycling and postmitotic cells from *Dcx*-expressing population subsets, *Mki67*- and *Ccnd2*-expressing cells (> 0.5 the log normalized count data) were considered SVZ cells whereas *Mki67*- and *Ccnd2*-negative cells (< 0.5 the log normalized count data) were labeled as MZ cells. To integrate multiple datasets, standard Seurat integration workflow was followed by performing the log normalization method. However, when combining previously reported datasets (GES103983 and GES109796; **Mayer et al., 2018**; **Mi et al., 2018**) with our data, the SCTransform normalization method was used under Seurat v3 integration workflow. For GSE109796, CGE, dMGE, and vMGE datasets from both E12.5 and E14.5 were analyzed. Cells that expressed the dorsal telencephalic cortical markers such as *Tbr1*, *Eomes*, *Neurod2,* and *Neurod6* were considered contamination from the microdissected GE samples and removed prior to SCTransform normalization.

## Fluorescent in situ hybridization and immunostaining

E12.5 and E14.5 embryonic brains were removed and drop-fixed in 4% paraformaldehyde overnight at 4°C. Fixed brains were washed in PBS, incubated in 30% sucrose overnight at 4°C and the cryopreserved. Tissues were cryosectioned at 14–16 µm in the coronal plane. RNAscope HiPlex

and HiPlexUp in situ hybridization assays (Advanced Cell Diagnostics) were performed according to the manufacturer's instructions. In situ hybridization images were taken using Zeiss AxioImager. M2 with or without ApoTome.2. Autofluorescence signals from blood vessels in embryonic brain tissues were corrected using Photoshop (Adobe) layer masking strategies. Image outlining was performed using ImageJ's (National Institutes of Health) Canny Edge Detector and superimpose of all images was conducted with RNAscope HiPlex Image Registration software (Advanced Cell Diagnostics).

For immunofluorescence, 16 µm sections were blocked for $\geq$ 1 hr at RT in blocking buffer (10% Normal Donkey Serum in PBS + 0.3% Triton X-100) and incubated at 4 °C overnight with primary antibodies in blocking buffer. Sections were washed 3 × 5 min at RT in PBS and incubated $\geq$ 1 hr at RT with fluorescent secondary antibodies and DAPI in blocking buffer. Before mounting, sections were washed again 3 × 5 at RT in PBS and images were captured with AxioImager.M2 with ApoTom.2. Primary antibodies used were rabbit anti-GFP (Invitrogen A11122, 1:400), chicken anti-doublecortin (Abcam AB153668, 1:1000) and mouse anti-Nestin (Millipore MAB353, 1:100).

## Acknowledgements

We thank Steven L Coon, James R Iben and Tianwei Li at the NICHD Molecular Genomics Core for sequencing services and initial Cell Ranger analyses; Dr. Hideyuki Okano for the Nestin-dVenus mice; and Dr. Dae-sung Kim and members of the Petros lab for helpful comments on the manuscript.

## Additional information

### Funding

| Funder | Grant reference number | Author |
| --- | --- | --- |
| Eunice Kennedy Shriver National Institute of Child Health and Human Development | Intramural Award | Timothy J Petros |

The funders had no role in study design, data collection and interpretation, or the decision to submit the work for publication.

### Author contributions

Dongjin R Lee, Conceptualization, Data curation, Formal analysis, Investigation, Methodology, Software, Validation, Visualization, Writing - original draft, Writing - review and editing; Christopher Rhodes, Data curation, Formal analysis, Methodology, Writing - review and editing; Apratim Mitra, Data curation, Formal analysis, Software, Writing - review and editing; Yajun Zhang, Dragan Maric, Investigation, Writing - review and editing; Ryan K Dale, Formal analysis, Software, Supervision, Visualization, Writing - review and editing; Timothy J Petros, Conceptualization, Investigation, Methodology, Project administration, Resources, Supervision, Visualization, Writing - original draft, Writing - review and editing

### Author ORCIDs

Dongjin R Lee (iD) http://orcid.org/0000-0002-3268-5687
Christopher Rhodes (iD) http://orcid.org/0000-0001-7438-4236
Timothy J Petros (iD) http://orcid.org/0000-0002-8943-546X

### Ethics

All mouse colonies were maintained in accordance with protocols approved by the Animal Care and Use Committee at the Eunice Kennedy Shriver National Institute of Child Health and Human Development (NICHD) under animal study protocol ASP #20-047.

### Decision letter and Author response

Decision letter https://doi.org/10.7554/eLife.71864.sa1
Author response https://doi.org/10.7554/eLife.71864.sa2

## Additional files

### Supplementary files

• Supplementary file 1. Differentially expressed genes in the VZ and SVZ. Table contains the genes that are enriched in the *Nestin*-positive VZ cells (1,083 genes), mitotic *Dcx*-positive SVZ cells (330 genes) and postmitotic *Dcx*-positive SVZ/MZ cells (1,320 genes), all with a minimum log-fold change = 0.25.

• Transparent reporting form

### Data availability

All of our sequencing data has been deposited in GEO under accession code GSE167013 and GSE190593.

The following datasets were generated:

| Author(s) | Year | Dataset title | Dataset URL | Database and Identifier |
|---|---|---|---|---|
| Lee DR, Rhodes CT, Mitra A, Zhang Y, Maric D, Dale RK, Petros TJ | 2021 | Transcriptional heterogeneity of ventricular zone cells throughout the embryonic mouse forebrain | https://www.ncbi.nlm.nih.gov/geo/query/acc.cgi?acc=GSE167013 | NCBI Gene Expression Omnibus, GSE167013 |
| Lee DR, Rhodes CT, Mitra A, Zhang Y, Maric D, Dale RK, Petros TJ | 2021 | Transcriptional heterogeneity of ventricular zone cells throughout the embryonic mouse forebrain | https://www.ncbi.nlm.nih.gov/geo/query/acc.cgi?acc=GSE190593 | NCBI Gene Expression Omnibus, GSE190593 |

The following previously published datasets were used:

| Author(s) | Year | Dataset title | Dataset URL | Database and Identifier |
|---|---|---|---|---|
| Mayer C, Hafemeister C, Bandler RC, Machold R, Batista Brito R, Jaglin X, Allaway K, Butler A, Fishell G, Satija R | 2018 | Developmental diversification of cortical inhibitory interneurons | https://www.ncbi.nlm.nih.gov/geo/query/acc.cgi?acc=GSE103983 | NCBI Gene Expression Omnibus, GSE103983 |
| Mi D, Li Z, Lim L, Li M, Moissidis M, Yang Y, Gao T, Hu TX, Pratt T, Price DJ, Sestan N, Marin O | 2018 | Early emergence of cortical interneuron diversity in the mouse embryo | https://www.ncbi.nlm.nih.gov/geo/query/acc.cgi?acc=GSE109796 | NCBI Gene Expression Omnibus, GSE109796 |

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
