## [Editor Report]

Your study is a thorough transcriptomics study demonstrating previously uncharacterized transcriptional diversity, discriminating the embryonic domains that produce interneurons. You have responded well to the requests for amendments, adjusted figures and better emphasized gene diversity in the ganglionic eminence.

---

## [Decision Letter]

**Decision letter after peer review:**

[Editors’ note: the authors submitted for reconsideration following the decision after peer review. What follows is the decision letter after the first round of review.]

Thank you for submitting the paper "Transcriptional heterogeneity of ventricular zone cells throughout the embryonic mouse forebrain" for consideration by *eLife*. Your article has been reviewed by 3 peer reviewers, and the evaluation has been overseen by a Reviewing Editor and a Senior Editor. The reviewers have opted to remain anonymous.

Comments to the Authors:

We are sorry to say that, after consultation with the reviewers, we have decided that this work will not be considered further for publication by *eLife* at this time.

The three reviewers found your study of transcriptional diversity of cortex and ganglionic eminences (GE) of the mouse embryo, focusing on progenitors that produce cortical interneurons of the brain, of importance to the field. They appreciated your use of a Nestin GFP line enabling enrichment for progenitors, and were positive about your finding of previously uncharacterized transcriptional diversity discriminating the embryonic domains producing interneurons.

Nonetheless, even with your novel findings and extensive analysis, the reviewers concurred that potential impact of the study is weakened by technical concerns on the sc RNA Seq analysis as well as by the lack of data supporting relevance of observed sub-types. They call for lineage analysis of at least one potential progenitor subtype to bolster the notion that transcriptionally heterogeneous progenitors are relevant to different fate outcomes. There was praise for the in situs but a call for additional in situ analysis, adding more genes and looking at different embryonic stages. Finally, they suggest that you pose your findings more broadly and beyond the lists of genes, classifying potential subtypes in a model to implicate which are primarily early born and those that are late born. Some of these revisions would be possible without performing additional work (cleaning up the sc RNA Seq analysis) but other inquiries would require further bench work. We hope that the reviewers' comments will help you to strengthen your story based on this data set.

*Reviewer #1:*

The manuscript describes the RNA fingerprint of developing telencephalic cells. To focus on progenitors, the authors use a transgenic line that drives the expression of a destabilized-venus GFP under the Nestin promoter. With this strategy, the authors find higher diversity of developing precursors than when they dissociate single cells from unlabeled WT brains because the latter contains higher proportions of postmitotic cells. The main finding is that early VZ precursors are more heterogeneous than inferred from previous studies. In situs in brain sections confirm molecularly distinct clusters of progenitors in medial, lateral, and dorsal VZ. The work is nicely written and organized. The results support the main conclusions. The findings advance the field, and the data will be useful in future research.

1. The work is nicely written and the conclusions about the heterogeneity of ventral precursors are well supported and important contribution to the field. However, the study of dorsal cell is minor. The analysis of dorsal cells seems to include many dorsal venus-expressing cells that are postmitotic neurons, and the number of analyzed cells from the dorsal origin is small.

My advice is to include the data of the study of dorsal cells as supplementary data, not as principal data.

2. On the other hand, I consider that the current supplementary Figure 1 showing the histological expression of the venus reporter should be presented as the main figure.

3. The field is rapidly offering new reports describing single cell analysis of telencephalic progenitors. The authors should revise and update the bibliography and discussion accordingly, including: "Single-cell transcriptomics of the early developing mouse cerebral cortex disentangle the spatial and temporal components of neuronal fate acquisition. Moreau et al., Development (2021) 148 (14): dev197962"; "Molecular logic of cellular diversification in the mouse cerebral cortex. Di Bella et la., Nature. 2021 Jul;595(7868):554-559".

*Reviewer #2:*

In this study, Lee et al. investigate the molecular diversity of cortex and ganglionic eminences (GE) of the mouse embryo. The authors used single-cell transcriptomics technology to acquire gene expression profiles of telencephalic regions, which they separated by microdissection of the lateral, caudal and medial ganglionic eminences along with the cortex in E12.5 and E14.5 mouse embryos. Using the Nes-dVenus line combined with Fluorescence-activated sorting they could enrich their dataset in ventricular zone cells, mainly neuronal progenitors. With this dataset, they could identify common differentiation lineages between the different GEs. Using the RNAscope technology, the authors validated unreported genes expressed in a specific ganglionic eminence and compartment within the GEs (VZ, SVZ, or MZ). In addition, the manuscript shows a bigger temporal mark on VZ cells compared to post-mitotic neurons between E12.5 and E14.5.

This manuscript presents a complete dataset of single-cell sequencing of ganglionic eminences during development. The effort put in the microdissection of the different GEs, central for the understanding of spatial implication in molecular identity and diversity, along with the validation of newly discovered genes with the RNAscope technique is very appreciated. Nevertheless, the manuscript shows significant weaknesses in the analysis of the transcriptomics data and quality controls. The overall tone is very descriptive such that this manuscript is rather intended for a specialized readership. I have the following comments:

1. In the methods, the authors describe the quality control applied to exclude bad quality cells from the data set (Supplementary Figure 2). At line 384, the authors say: "we first removed cells that had unique molecular identifies (UMI) counts over 4500 or less than 200." From what is shown in supplementary figure 2 we observe that this filtering is done on the number of expressed genes and not on the UMI counts. In addition, in the violin plot of a number of genes per cell, the distribution looks binomial with a population of cells expressing around 500 genes (highlighted in orange) and another population expressing around 3000 genes per cell (highlighted in blue below). The population with a low number of genes expressed should be excluded for further analysis because it is likely that it represents 10x GEMs containing ambient RNA or dying cells instead of GEMS containing a whole cell. This is supported by the post-filtering graph at the bottom of the figure (% of Mitochondrion vs Counts Depth) where, strikingly for the cortex, we observe a population of cells with low % of mitochondrial RNA and high Count depth and a band of cells formed by low Count Depth cells with a variety of mitochondrial RNA (highlighted in red). I strongly recommend the authors filter out the low number of genes population (in orange) by setting a threshold of around 1500 genes per cell which will probably clean the populations with low count depth and thus avoid including bad quality GEMs for further analysis.

2. At the lines 90-92 of the manuscript, the authors write: "Conversely, cortical neurogenesis does not begin until ~E14 (Noctor et al., 2004; Sessa et al., 2008; Tyler et al., 2015), so the majority of E12.5 cortical cells are mainly VZ neural progenitors, specifically RGCs." Indeed, at E12.5 the ventricular zone is the predominant compartment of the developing pallium and most of the cells present at this stage are radial glial cells, nevertheless, asymmetric division of radial glia to produce neurons has been reported as soon as E11.5 in the mouse (Kwan et al. Development, 2012, doi: 10.1242/dev.069963), explaining the small but present proportion of neurons found in the dataset.

3. In line 258 the authors write: "To compare the E12.5 and E14.5 cells, we integrated these datasets together". It is unclear whether the two datasets were just merged in a common Seurat object or if they were integrated with SCTransform normalization and integrated with the Seurat v3 pipeline. I would suggest that this sort of integration not be applied when combining datasets generated in such similar conditions because it can remove the differences between datasets, in this case, age of collection.

4. In Figure 7, the authors nicely show a big overlap and few differentially expressed genes in post-mitotic cells between E12.5 and E14.5. It would be very interesting to know if this overlap is shared for different areas of the telencephalon such as the pallium or if it is specific to the subpallium.

5. In order to isolate basal progenitor cells, the authors selected cells expressing both neuronal markers such as Dcx and cycling marker as Mki67. Since in the 10x experiments it is frequent to have doublets and their exclusion is usually very hard in standard quality control based on genes and UMI counts, I would recommend running a doublet prediction analysis (as DoubletFinder McGinnis et al., 2019 doi:10.1016/j.cels.2019.03.003 ), in order to be sure that the population isolated are indeed basal progenitors and not doublets composed of progenitor and neuron.

6. The sequencing of spatially resolved data is extremely interesting and opens the possibility for addressing the contribution of space to the transcriptome. Here it would be extremely interesting to examine whether the different ganglionic eminences have sharp distinct molecular identities or if there is a gradient of identity between the different eminences.

7. From line 187 on, the supplementary figures references are wrongly numbered.

*Reviewer #3:*

This study investigates transcriptional diversity of cortical interneuron progenitors in embryonic mice. A major strength of this study is the use of a Nestin GFP line which enables the authors to enrich for progenitors, more so than has been done in previous studies. Using this line the authors uncover previously uncharacterized transcriptional diversity discriminating embryonic domains which produce interneurons at an early stage of development (E12.5). Importantly, the authors make use of smFISH to validate many of these changes. The study provides new lists of transcripts which may highlight diversity of progenitors across domains. Finally the authors perform some analysis at E14.5 to evaluate temporal differences.

While the study has some important novelty and generates evidence of transcriptional diversity, the potential impact of this study is weakened by technical concerns as well as lack of data supporting relevance of observed sub-types. Lineage analysis of at least one potential progenitor subtype would further support the notion that transcriptional heterogeneous progenitors are important (give rise to different fates) as opposed to transcriptional noise.

1. A major conclusion of this paper is that progenitors exhibit transcriptional heterogeneity. When focusing specifically on VZ cells with high Nestin expression (figure 4), how many cells are the authors examining-ie how many cells is this conclusion based upon? Likewise, it would be helpful to better understand how many samples were sequenced to conclude heterogeneity as well as how replicates compare. From the methods it appears that E12.5 is based on 2 Nestin GFP mice. This raises the question as to what extent is this transcriptional diversity reflecting noise of different samples (which could be slightly different stages?) or even technical variation of sequencing.

2. Previous studies have used scSEQ of GE at E13.5 and E14.5 (Fishell lab) and E12.5-E14.5 (Marin lab). The discriminating point here is that the current study uses FACS and a Nestin reporter to isolate and sequence specifically VZ cells. The previous studies also examined VZ cells within their sequenced population, with the Marin study highlighting heterogeneous progenitors evident at E12.5 as well as across MGE and CGE. In the current study the authors conclude their gene expression profiles provide new insight into transcriptional diversity of interneurons. However mainly this study is a list of differential genes (albeit which extends that previously identified). To make this study more impactful it would be helpful if the authors could provide some data such as lineage analysis to support the conclusion that observed diversity impacts fate.

3. Further classification of potential subtypes in a model of some kind would also be helpful-ie are there subtypes which are primarily early born and others that are late born, etc?

4. The authors use Nestin GFP to sort out VZ cells in this study. In Figure 1 they show that Nestin- GFP cells also express markers of newborn neurons (Dcx positive), neurons (Tbr1), and IPS/neurons (Neurod6), particularly in the cortex. The authors should clarify if this is reflecting some perdurance of the GFP into post-mitotic cells? As figure 1 includes WT cells, it is unclear in subsequent figures (such as Figure 3) if they are only focusing on GFP sorted cells or if they are segregating data solely by looking at Nestin transcript. This may be a simple text fix but these missing details make the paper confusing. Due to potential perdurance of GFP protein and even transcript these are not necessarily the same.

5. Analysis of E14.5 is limited-especially as the authors again just sample 1 nestin GFP brain at this stage. The authors conclude that there is more diverse neurogenic and postmitotic expression at E14.5. Additional analysis of cell populations as well as validation would be important to tease out these differences and support this conclusion.

[Editors’ note: further revisions were suggested prior to acceptance, as described below.]

Thank you for resubmitting your work entitled "Transcriptional heterogeneity of ventricular zone cells throughout the embryonic mouse forebrain" for further consideration by *eLife*. Your revised article has been evaluated by Marianne Bronner (Senior Editor) and a Reviewing Editor.

The manuscript has been improved but there are some remaining issues that need to be addressed, as outlined below:

The reviewers find your paper valuable and suggest several revisions that should improve the presentation of your study. Although Reviewer #1 appears satisfied with the study in the appended review, in the Reviewer Consultations offline, this reviewer agreed with the other two that a deeper analysis of the expression (in situ and immunostaining) at different developmental times (at least the 2 stages you sequenced) would verify whether a specific marker is stable or not in CGE MGE and LGE. Fabp7 characterization in particular would be welcome; if the results show selectivity, you could minimally address the request of Reviewer 4 to highlight a marker.

This reviewer would also like to revisit the previous request to move data concerning the dorsal progenitors to supplementary data so as to further focus on the GE, to emphasize the ganglionic eminences.

Reviewer 2 asks for highlighting the impact of the study with respect to the ganglionic eminences, to better showcase the special focus of your work. This can be done by modifying the summary to better represent content of the paper.

Also recommended is to address in the Discussion the discordance between spatial (in situ) patterns vs sc seq for Id4 and Mest.

Finally, and readily done, is to place in situs next to the relevant UMAPs in the figures; this would aid the reader's navigation through your very detailed and rich data.

Reviewer 4 is new and asks for proof of some genes suggested to be region-specific, such as Igfbp5, and also linking progenitor heterogeneity to proliferation or fate differences or even protein levels, toward using these differences to isolate subsets by FACS. Proof that expression differences link to functional differences, by, for example, targeting candidate genes with a Cre construct, or doing localization/birthdating. Targeting genes experimentally would involve much more extensive work at this point, and Reviewer 1 especially agrees. But all three reviewers believe that because your paper is very descriptive, if you could add some functional evidence, it would increase the paper's impact. At the very least, you could try to do birthdating/cell cycle analyses, and should they work, this would help the significance of the paper. Minimally, the deeper analysis of Fabp7 expression is warranted, and should address the "ask" of Reviewer 4 to provide a marker for further cell isolation.

*Reviewer #1:*

The authors have resolved the main concerns of the manuscript. I consider that the addition of the new controls improves the quality of the manuscript. The figures showing comparisons with previous studies explain the added value of this study.

*Reviewer #3:*

The authors have been generally attentive to concerns raised. They have adjusted their analysis of scSEQ datasets, included new validation experiments for E14.5, and made some additional minor changes. They have revised the writing of the manuscript to explain the novelty of their findings relative to other papers (one of my original concerns) as well as to flesh out more details which increases readability. While the paper is still largely descriptive, the authors have generated some valuable new insights for the field. I do have a few remaining minor suggestions.

1. The impact of this study is mainly their analysis of the ganglionic eminences where interneurons are born. In this regard the authors might consider adjusting the summary to reflect that emphasis even more. This may attract more readers to their work and increase impact.

2. The finding that Id4 and Mest spatial patterns by in situ are different than that predicted from the scseq is interesting (but also somewhat concerning for the field). It would be nice if the authors comment on this discordance in the discussion as an important point of consideration for sc seq experiments.

3. Figure 7-supplement 1F labels Nestin expressing cells. The use of red for labeling nestin is confusing here as red also labels CGE.

4. The representation of the in situs adjacent to UMAPS in Figure 8 is really helpful. For this figure I think they should also add in annotations with the UMAPs labeled for each respective region/stage. There are labels for panel A but not the plots below. In addition, I wish the authors would use this approach in prior in situ figures-ie include the relevant UMAPs beside the in situs (for perhaps a subset), as opposed to putting in supplement. It facilitates analysis of their data.

5. Line 366 in discussion (types may be singular)

*Reviewer #4:*

In this manuscript the authors profile nestin-expressing cells from the 3 ganglionic eminences in E12 and E14 murine embryos by scRNA-seq. This yields several region-specific genes, some of which could serve as a new regional marker, such as Igfbp5 for CGE, but they do not prove this. Most importantly, they do not analyze the region-specific differences across at least the 2 stages they sequenced to verify if these markers are stable at least for these 2 days in development.

I fully agree with the authors that it is important to explore progenitor subtypes at higher resolution, i.e. many cells, to explore heterogeneity. This is also highly relevant for the diverse output from these regions, including the generation of adult neural stem cells. However, the manuscript falls short of linking either progenitor heterogeneity or regional heterogeneity to a specific function – e.g. proliferation differences, fate differences or protein levels to allow subset isolation by FACS. Please provide at least 1 example to showcase that the expression differences link to functional differences in output (in utero-electroporation of a Cre construct targeted to a specific subset) or highlighting by RNAscope and EdU cell cycle monitoring that the marker population has indeed specific cell cycle properties (slow division, for example).

One specific example that raises doubts about the usefulness of the markers proposed is Fabp7, as indeed staining seems to label all cells in the ventricular zone, even though levels are always higher at boundary regions. This would mean that the "absent" expression of Fabp7 may simply be due to the depth of sequencing. Therefore, I suggest the authors perform immunostainings to verify or falsify the usefulness of this "marker".

This leads to my last suggestion namely to focus the analysis to some extent on genes encoding cell surface proteins to allow sorting subsets of progenitors.

---

## [Author Response]

[Editors’ note: The authors appealed the original decision. What follows is the authors’ response to the first round of review.]

Comments to the Authors:Reviewer #1:The manuscript describes the RNA fingerprint of developing telencephalic cells. To focus on progenitors, the authors use a transgenic line that drives the expression of a destabilized-venus GFP under the Nestin promoter. With this strategy, the authors find higher diversity of developing precursors than when they dissociate single cells from unlabeled WT brains because the latter contains higher proportions of postmitotic cells. The main finding is that early VZ precursors are more heterogeneous than inferred from previous studies. In situs in brain sections confirm molecularly distinct clusters of progenitors in medial, lateral, and dorsal VZ. The work is nicely written and organized. The results support the main conclusions. The findings advance the field, and the data will be useful in future research.

We thank the reviewer for their positive comments on our manuscript.

1. The work is nicely written and the conclusions about the heterogeneity of ventral precursors are well supported and important contribution to the field. However, the study of dorsal cell is minor. The analysis of dorsal cells seems to include many dorsal venus-expressing cells that are postmitotic neurons, and the number of analyzed cells from the dorsal origin is small.My advice is to include the data of the study of dorsal cells as supplementary data, not as principal data.

The reviewer is correct that our main focus is on the ventral telencephalon, and all figures and analysis after Figure 1 are focused on the GEs. However, we feel that presenting all of the dorsal and ventral sequenced cells together in one introductory figure is the most comprehensive way to display our entire dataset, and thus we prefer to keep it in the main text. This approach allows us to compare pan-excitatory vs. inhibitory cell markers to verify the quality of the dataset, as in Figure 1C-D. And it’s unclear how useful it would be to segregate the dorsal cells into their own supplementary figure, as it removes the benefit of comparing/contrasting to the GE dataset. For these reasons, we prefer to keep the dorsal cells in Figure 1 and not move extract them into a supplement figure.

Also, one point of clarification: neurogenesis in the dorsal cortex begins later that in the GEs with only layer VI cortical pyramidal cells beginning to be born around E12.5 (please see comments to Reviewer #2 below). Thus, the majority of all E12.5 dorsal cortical cells (and especially Venus-expressing cells) are still cycling progenitors, likely ~80% based on Di Bella…Arlotta *Nature* 2021; there are very few dorsal cortex postmitotic neurons at E12.5.

2. On the other hand, I consider that the current supplementary Figure 1 showing the histological expression of the venus reporter should be presented as the main figure.

We have moved the immunostaining of the Nes-dVenus mouse from Figure 1—figure supplement 1 into the main text as new Figure 1A. Now Figure 1—figure supplement 1 is entirely devoted to the flow cytometry gating strategy and plots.

3. The field is rapidly offering new reports describing single cell analysis of telencephalic progenitors. The authors should revise and update the bibliography and discussion accordingly, including: "Single-cell transcriptomics of the early developing mouse cerebral cortex disentangle the spatial and temporal components of neuronal fate acquisition. Moreau et al., Development (2021) 148 (14): dev197962"; "Molecular logic of cellular diversification in the mouse cerebral cortex. Di Bella et la., Nature. 2021 Jul;595(7868):554-559".

We thank the reviewer for pointing out these relevant references and have added them (and several others) in the introduction of the manuscript on p. 3, lines 59-62.

Reviewer #2:In this study, Lee et al. investigate the molecular diversity of cortex and ganglionic eminences (GE) of the mouse embryo. The authors used single-cell transcriptomics technology to acquire gene expression profiles of telencephalic regions, which they separated by microdissection of the lateral, caudal and medial ganglionic eminences along with the cortex in E12.5 and E14.5 mouse embryos. Using the Nes-dVenus line combined with Fluorescence-activated sorting they could enrich their dataset in ventricular zone cells, mainly neuronal progenitors. With this dataset, they could identify common differentiation lineages between the different GEs. Using the RNAscope technology, the authors validated unreported genes expressed in a specific ganglionic eminence and compartment within the GEs (VZ, SVZ, or MZ). In addition, the manuscript shows a bigger temporal mark on VZ cells compared to post-mitotic neurons between E12.5 and E14.5.This manuscript presents a complete dataset of single-cell sequencing of ganglionic eminences during development. The effort put in the microdissection of the different GEs, central for the understanding of spatial implication in molecular identity and diversity, along with the validation of newly discovered genes with the RNAscope technique is very appreciated. Nevertheless, the manuscript shows significant weaknesses in the analysis of the transcriptomics data and quality controls. The overall tone is very descriptive such that this manuscript is rather intended for a specialized readership. I have the following comments:1. In the methods, the authors describe the quality control applied to exclude bad quality cells from the data set (Supplementary Figure 2). At line 384, the authors say: "we first removed cells that had unique molecular identifies (UMI) counts over 4500 or less than 200." From what is shown in supplementary figure 2 we observe that this filtering is done on the number of expressed genes and not on the UMI counts. In addition, in the violin plot of a number of genes per cell, the distribution looks binomial with a population of cells expressing around 500 genes (highlighted in orange) and another population expressing around 3000 genes per cell (highlighted in blue below). The population with a low number of genes expressed should be excluded for further analysis because it is likely that it represents 10x GEMs containing ambient RNA or dying cells instead of GEMS containing a whole cell. This is supported by the post-filtering graph at the bottom of the figure (% of Mitochondrion vs Counts Depth) where, strikingly for the cortex, we observe a population of cells with low % of mitochondrial RNA and high Count depth and a band of cells formed by low Count Depth cells with a variety of mitochondrial RNA (highlighted in red). I strongly recommend the authors filter out the low number of genes population (in orange) by setting a threshold of around 1500 genes per cell which will probably clean the populations with low count depth and thus avoid including bad quality GEMs for further analysis.

We thank the reviewer for carefully exploring our dataset and pointing out the bimodal distributions within our graphs in Figure 1—figure supplement 2. With our goal of maximizing cell numbers, it’s possible we under-filtered our dataset. ~84,000 E12.5 cells successfully passed through the Cell Ranger pipeline (which was depicted in the chart in old Figure 1). Then the combination of removing cells with > 4500 or < 200 genes per cell, cells with high mitochondria transcripts, and cells with hemoglobin subunit *Hbb* from WT mice (all described in the original Methods) resulted in ~43,000 cells being analyzed in the original manuscript.

As the reviewer suggested, we increased the threshold for number of genes from > 200 to > 1,500 per cell to filter out potential ambient RNA, dying cells, etc. with low count depth that the reviewer noted (Figure 1—figure supplement 3b). (Relatedly, both DAPI and DRAQ5 were added to the single cell suspension prior to flow cytometry so that we could collect live cells (DRAQ5^+^/DAPI^-^) and remove dead/dying cells (DRAQ5^+^/DAPI^+^), so there should be very few dead/dying cells in our analysis.) We also removed those predicted doublets that had *DoubletFinder* scores greater than 0.3 from further downstream analysis (see point #5 below). Using the > 1,500 gene/cell cutoff, this reduced our final counts from ~43,000 E12.5 cells to ~36,500 E12.5 cells. (We performed identical analysis on the E14.5 population in Figure 7—figure supplement 1, which resulted in ~24,000 high quality E14.5 cells). As you can see from the figure below, we still observe a clean separation of the CTX and MGE cells whereas the LGE and CGE populations are still largely intermingled; the new dotplot looks very similar to the original plot (Author response image 1 and Figure 1—figure supplement 3a). Furthermore, the top 20 DEGs between the regions are very similar to our original analysis (Figure 1—figure supplement 3b). Thus, we strongly believe that no significant biological misinterpretation was made using the previous > 200 gene/cell cutoff, and that possible contamination from ambient mRNA or dead/dying cell datapoints did not affect any data interpretation or conclusions from our dataset. This notion is strengthened by the fact that the overwhelming majority of genes characterized by our in situ hybridizations were in agreement with their predicted expression profiles from the scRNAseq data.

**Author response image 1. sa2fig1:** 

Using the more stringent > 1,500 gene per cell threshold with the ~36,500 E12.5 cells, we have updated the dotplots and heatmaps in Figures 1 and 2, as well as related Figure 1—figure supplement 3 and Figure 2—figure supplement 3. We have also updated Figure 1—figure supplement 2B and Figure 7—figure supplement 1 using the > 1,500 gene per cell threshold.Note that these low genes per cell datapoints were already excluded from the Nestin- and Dcx-enriched datasets in Figures 3, 4, 7, Figure 6—figure supplement 1 and related analysis because they did not reach the gene expression threshold anyway; thus, this portion of our analysis in this manuscript was unaffected by this change in gene per cell threshold.

Again, we thank the reviewer for this observation and now have more confidence in the quality of cells in our dataset. And importantly, there are no significant changes in the overall conclusions and gene expression profiles originally detected.

2. At the lines 90-92 of the manuscript, the authors write: "Conversely, cortical neurogenesis does not begin until ~E14 (Noctor et al., 2004; Sessa et al., 2008; Tyler et al., 2015), so the majority of E12.5 cortical cells are mainly VZ neural progenitors, specifically RGCs." Indeed, at E12.5 the ventricular zone is the predominant compartment of the developing pallium and most of the cells present at this stage are radial glial cells, nevertheless, asymmetric division of radial glia to produce neurons has been reported as soon as E11.5 in the mouse (Kwan et al. Development, 2012, doi: 10.1242/dev.069963), explaining the small but present proportion of neurons found in the dataset.

The reviewer is correct that there is growing evidence that the earliest cortical neurons (layer VI) are born prior to E14, likely starting to be generated ~E11.5-E12. My lab has performed EdU labeling at E11.5 and found VERY few cortical neurons that are EdU-labeled in the adult, but we do see subplate cells that are EdU^+^. And based on the recent Di Bella…Arlotta 2021 *Nature* paper, it appears that ~80% of E12.5 dorsal cortical cells are classified as APs or intermediate progenitors whereas < 5% of cells are classified as ‘immature neurons’ (see Extended Data Figure 1). Based on these findings, we have modified the sentence to more accurately reflect the timing of cortical development, p.4 line 94-96: “Conversely, the vast majority of E12.5 dorsal cortical cells are VZ neural progenitors (specifically RGCs), with a small number of layer VI projection neurons starting to emerge at this time (Kwan et al., 2012; Di Bella et al., 2021).”

3. In line 258 the authors write: "To compare the E12.5 and E14.5 cells, we integrated these datasets together". It is unclear whether the two datasets were just merged in a common Seurat object or if they were integrated with SCTransform normalization and integrated with the Seurat v3 pipeline. I would suggest that this sort of integration not be applied when combining datasets generated in such similar conditions because it can remove the differences between datasets, in this case, age of collection.

We apologize for not being clearer in the main text and Methods about how we merged these two datasets. For the dataset comparing the E12.5 and E14.5 cells in Figure 7, standard Seurat integration workflow was followed performing the log normalization method as opposed to *SCTransform*, using E12.5 and E14.5 as a *reference list*. To clarify this point, we have updated the Methods section with more details about the normalization methods we used for each analysis in the ‘Single-cell RNA sequencing Analysis’ section (p. 18-19).

4. In Figure 7, the authors nicely show a big overlap and few differentially expressed genes in post-mitotic cells between E12.5 and E14.5. It would be very interesting to know if this overlap is shared for different areas of the telencephalon such as the pallium or if it is specific to the subpallium.

We did not generate scRNAseq experiments on the E14.5 pallium/CTX, so we are unable to perform the requested comparison. As the reviewer accurately noted in the above comment, there is a very small number of postmitotic cortical cells at E12.5, while there would be a significant expansion of postmitotic cells at E14.5. I do not know if our dataset contains enough postmitotic E12.5 cortical cells to definitely state one way or another if this overlap would be subpallium-specific or if it applied more generally to other brain regions. Since the cortex is producing primarily layer VI cells at E12.5 and layer III-IV cells at E14.5, I would not be surprised if there were more distinct genetic profiles of postmitotic cells at these ages. There are more comprehensive dorsal cortex datasets available to better answer this specific question, such as the one generated in the Di Bella…Arlotta 2021 *Nature* paper.

5. In order to isolate basal progenitor cells, the authors selected cells expressing both neuronal markers such as Dcx and cycling marker as Mki67. Since in the 10x experiments it is frequent to have doublets and their exclusion is usually very hard in standard quality control based on genes and UMI counts, I would recommend running a doublet prediction analysis (as DoubletFinder McGinnis et al., 2019 doi:10.1016/j.cels.2019.03.003 ), in order to be sure that the population isolated are indeed basal progenitors and not doublets composed of progenitor and neuron.

We thank the reviewer for pointing out the limitation of 10X experiments with regards to dealing with doublets. Seurat attempts to identify and filter out suspected doublets with a clear outlier number of UMIs and genes detected, which we have incorporated to our analyses. However, this pipeline is not perfect in identifying and removing doublets. As the reviewer suggested, we employed the doublet prediction algorithm *DoubletFinder* to identify possible doublets in our dataset (scores greater than 0.3). The resulting *DoubletFinder* plot is shown in author response image 2. As you can see in Author response image 2, *DoubletFinder* identified a group of cells in the middle of the plot (cluster #11) as high potential doublets. Notably, this was also the only cluster that had a pretty even mix of MGE/CGE/LGE-derived cells. Importantly, there was only a smattering of potential doublets scattered amongst the other clusters, indicating that there is little reason for believing that there are any potential doublet issues with the overwhelming majority of cells outside of cluster #11. Thus, there is no evidence that the basal progenitors are ‘false positives’ consisting of a progenitor and neuron doublet.

Based on this observation, we reran the genetic analysis in Figure 3 and all subsequent figures after removing predicted doublets. We have generated new dotplots with these predicted doublets removed for Figures 3, 4, 7 and Figure 5—figure supplement 1, Figure 6—figure supplement 1, Figure 6—figure supplement 2 and Figure 7—figure supplement 2. We also generated new heatmaps in Figures 3, 4, 7 and Figure 6—figure supplement 1 after removal of these predicted doublets. Importantly, these adjustments did not generate any significant changes in the data. In some cases, the gene order may have jumped around a little (e.g., the second most enriched gene is now the fifth, or vice versa). All conclusions we originally made are still supported by the data, which is expected as we only removed ~4% of all cells/doublets that likely had minimal influence in the overall genetic comparisons between regions, cell types and time points. We thank the reviewer for bringing *Doubletfinder* to our attention to help clean up our data, and we plan to use it for future experiments to ensure that we remove all predicted doublets from our dataset. We have updated the Results text on p. 5 and Methods text in the ‘Single-cell RNA Sequencing Analysis’ section on p. 18-19 to reflect these changes.

6. The sequencing of spatially resolved data is extremely interesting and opens the possibility for addressing the contribution of space to the transcriptome. Here it would be extremely interesting to examine whether the different ganglionic eminences have sharp distinct molecular identities or if there is a gradient of identity between the different eminences.

Based on our in situ data, we see both types of expression patterns: some genes display sharp, distinct boundaries that seem to adhere to the GE boundaries, such as *Mt3* and *Rbp1* in the MGE/LGE boundary (Figure 5G). Other genes display gradient expression profiles that may blur or ignore the GE boundaries such as *Id4* and *Nkx6.2* at the MGE/LGE boundary (Figure 5F, H). Some genes even display sharp boundaries within a specific GE, such as *Rbp1* and *Igfbp5* in the dCGE and vCGE, respectively (Figure 5G). The idea of sharp molecular boundaries vs. gradients of gene expression has important implications in gene function throughout developmental biology, with gradients of various signaling factors playing critical roles in initial patterning and fate determination throughout the early stages of neurogenesis. We agree with the reviewer that these patterns are extremely interesting, and the function of various genes in specific brain regions with either gradient and/or sharp boundaries warrant future exploration. We have added a sentence in the discussion noting the observation of both sharp boundaries and gradients, p. 14 lines 338-339.

7. From line 187 on, the supplementary figures references are wrongly numbered.

Thank you for pointing this out, we apologize for these errors and have corrected the Supplement Figure numbers in the revised submission.

Reviewer #3:This study investigates transcriptional diversity of cortical interneuron progenitors in embryonic mice. A major strength of this study is the use of a Nestin GFP line which enables the authors to enrich for progenitors, more so than has been done in previous studies. Using this line the authors uncover previously uncharacterized transcriptional diversity discriminating embryonic domains which produce interneurons at an early stage of development (E12.5). Importantly, the authors make use of smFISH to validate many of these changes. The study provides new lists of transcripts which may highlight diversity of progenitors across domains. Finally the authors perform some analysis at E14.5 to evaluate temporal differences.While the study has some important novelty and generates evidence of transcriptional diversity, the potential impact of this study is weakened by technical concerns as well as lack of data supporting relevance of observed sub-types. Lineage analysis of at least one potential progenitor subtype would further support the notion that transcriptional heterogeneous progenitors are important (give rise to different fates) as opposed to transcriptional noise.

Please see responses to Reviewer #2 above and specific comments below regarding the technical concerns. And please see our response to point #2 below regarding lineage analysis.

1. A major conclusion of this paper is that progenitors exhibit transcriptional heterogeneity. When focusing specifically on VZ cells with high Nestin expression (figure 4), how many cells are the authors examining-ie how many cells is this conclusion based upon? Likewise, it would be helpful to better understand how many samples were sequenced to conclude heterogeneity as well as how replicates compare. From the methods it appears that E12.5 is based on 2 Nestin GFP mice. This raises the question as to what extent is this transcriptional diversity reflecting noise of different samples (which could be slightly different stages?) or even technical variation of sequencing.

To answer the reviewer’s first point, the number of high Nestin-expressing cells from Figure 4 is 9,308 (with 3036 from LGE, 3890 from MGE and 2382 from CGE). This is the number reported in Sup Figure 5 where we compare our study to the Fishell & Marin studies. Similarly, the number of high DCX-expressing cells from Sup Figure 9 is 12,307 (3900 LGE, 3902 MGE, 4505 CGE). We have included these cell numbers in the main text on lines p.8 line 194-95 and p.10 lines 243-44.

We noted the total number of sequenced cells and replicates in the Methods, but it certainly could have been presented more clearly. In total, our manuscript analyzed ~60,000 high quality E12.5 and E14.5 cells which can be broken down as follows: (A) 3 scRNAseq reactions for each E12.5 MGE/LGE/CGE (1 WT and 2 Nes-dVenus) resulting in 9,275-12,052 cells per GE, (B) 2 scRNAseq reactions for E12.5 cortex (1 WT and 1 Nes-dVenus) resulting in ~4,785 CTX cells, and (C) 2 scRNAseq reactions for each E14.5 MGE/LGE/CGE (1 WT and 1 Nes-dVenus) resulting in 7,171-8,658 cells per GE. For each reaction, tissue from a minimum of 4 embryos were combined. We have reworded the Methods section to more clearly state this breakdown of cell numbers in the ‘Single-cell RNA sequencing Analysis’ section, p. 19.

Author response image 3 displays the 3 E12.5 replicates (1 WT and 2 Nes-dVenus) overlaid on top of each other: on the left are the 3 GEs combined (comparable to Figure 2A in the manuscript), and on the right is the MGE extracted out (comparable to Figure 2B in the manuscript). As you can see, the replicates have a high degree of overlap, with very little evidence for diversity being due to differences in replicates or technical variation of sequencing. The main difference is the bias for WT-derived cells to be on the right side of the figure (DCX^+^-region) and Nes-dVenus on the left side (Nes^+^-region), as expected. Similar results are observed when extracting out the LGE and CGE.

**Author response image 3. sa2fig3:** 

For comparison to other recent high-quality single cell sequencing papers of embryonic mouse forebrain: (1) Di Bella…Arlotta 2021 *Nature* paper performed a comprehensive scRNAseq analysis of mouse embryonic dorsal cortex every day from E10.5-E17.5. This study consisted of 1 replicate on all cortical embryonic timepoints from E10.5-E17.5, with a range of ~3,000-11,600 cells per timepoint (mean ~7,000/timepoint). (2) Moreau…Causeret *Development* 2021 paper analyzed a total of 4,225 cells from 1 replicate near the pallial/subpallial boundary. (3) Telley…Jabaudon *Science* 2019 paper analyzed a total of 2,756 cortical cells spread across 12 different embryonic cell labeling and collection conditions, 1 replicate each, which averages out to ~230 cells/timepoint. (4) Preissel…Ren *Nat Neuro* 2018 performed scATACseq on ~12.7K cells from 7 different timepoints ranging from E11.5 to P0, with 2 replicates/timepoint, which averages out to ~900 cells per replicate (or ~1800/timepoint, see Sup Table 2 in this paper). (5) Mi…Marin *Nature* 2018 paper sequenced a total of 2,000 cells from 3 different brain regions at 2 different embryonic timepoints (1 replicate each), which averages out to ~330 cells/timepoint. (6) Mayer…Fishell *Nature* 2018 paper performed scRNAseq on a total of 5,600-8,500 cells from the MGE, LGE and CGE, with multiple replicates of each region.Thus, we believe that our significant increase cell number (and thus power) compared to many other single cell sequencing studies, combined with our specific enrichment of VZ cells with the Nes-dVenus mouse, is why we detect many novel gene expression profiles and candidate progenitor subgroups that could not have been detected in other studies. Especially in regards to VZ cells/RGCs which are underrepresented in these other studies.

2. Previous studies have used scSEQ of GE at E13.5 and E14.5 (Fishell lab) and E12.5-E14.5 (Marin lab). The discriminating point here is that the current study uses FACS and a Nestin reporter to isolate and sequence specifically VZ cells. The previous studies also examined VZ cells within their sequenced population, with the Marin study highlighting heterogeneous progenitors evident at E12.5 as well as across MGE and CGE. In the current study the authors conclude their gene expression profiles provide new insight into transcriptional diversity of interneurons. However mainly this study is a list of differential genes (albeit which extends that previously identified). To make this study more impactful it would be helpful if the authors could provide some data such as lineage analysis to support the conclusion that observed diversity impacts fate.

The reviewer is correct that the previous studies from the Fishell (Mayer…Fishell 2018) and Marin lab (Mi…Marin 2018) performed scRNAseq of the mouse ventral telencephalon. A couple of notes on the Marin paper: First, as stated above, they sequenced a total of 2,003 cells from the vMGE, dMGE and CGE at E12.5 and E14.5 (six different data points), around half of which were classified as ‘progenitors’ (VZ and SVZ cells). This averages out to ~150 progenitor cells per region per age, which is significantly underpowered for making strong conclusions about distinct VZ and SVZ progenitor cell types. And based on our analysis, the entire Marin dataset contains only 147 high Nestin expressing/VZ cells (our Sup Figure 5). Additionally, they do not perform any confirmation of these gene expression profiles (in situs, immunostaining, etc.).

Comparatively, our dataset averages ~9,300 cells per GE per age, with the vast majority of collected cells being VZ or SVZ progenitors because of using the Nes-dVenus mouse. And our in situs confirm spatially and temporally restricted expression patterns of numerous genes from the scRNAseq dataset. For these reasons, we argue that our dataset does more than ‘…extend previously identified [genes]’. The Marin study was significantly underpowered to make any definitive claims about differential gene expression at these timepoints, especially regarding cycling VZ and SVZ cells. And the Fishell study does not closely explore these cycling cell populations; their focus is more on the developmental progression of postmitotic MGE cells. Thus, we argue that our study has uncovered an entirely new genetic organization and diversity in the embryonic GEs, and particularly in cycling VZ progenitors, that has not been previously identified. To our knowledge, the majority of gene expression patterns confirmed by our in situs are novel.

The author raises in interesting and important point about linking gene expression patterns to specific GABAergic cell fates. This is an important and challenging question to tackle throughout developmental biology. We are confident that some of these genes we characterize here likely play important roles in fate determination of specific interneuron subtypes. For example, it has been shown that there is a bias of SST^+^ cells born from the dorsal MGE and PV^+^ cells born from the ventral MGE (Wonders…Anderson 2008; Sousa…Fishell 2009; Inan…Anderson 2012). In particular, there is a strong bias for Nkx6.2 lineage cells from the dMGE to generate SST^+^ cells (Figure 5H; Sousa…Fishell 2009). However, almost nothing is known about the relationship between spatial domains and cell fate in the LGE and CGE.

For the genes with the most intriguing, spatially refined expression profiles (such as *Igfbp5*, *Wnt7b Rmst* and *Cxcl14* in the CGE), we searched the literature for Cre-drivers that would allow us to fate-map these populations in the CGE, but could not find live mice with these drivers. We also did not find live floxed alleles for the relevant genes of interest that would allow us to study their function. A *Cxcl14* floxed mouse was generated in 2010, but it’s unclear if this line is still maintained as a live colony. Plus, the lack of a specific CGE-Cre driver makes this approach challenging. In utero electroporation, which our lab has extensive experience with, do not provide the necessary spatial targeting resolution to label or manipulate subdomains of GEs. Thus, without proper genetic tools, it is extremely challenging (if not impossible) to definitively target specific subpopulations of the mouse GEs and correlate these populations with mature interneuron fate. We are planning on generating Cre-drivers for 1-2 of the most intriguing candidates we have uncovered, but this will take ~6-12 months to generate the mice, then cross them to appropriate reporters, etc.; this is a much longer-term project. Therefore, while we agree with the reviewer’s interest in lineage tracing, this is beyond the scope of the current manuscript because we currently lack the genetic tools to properly characterize embryonic gene expression with mature cell fate in a timely manner.

3. Further classification of potential subtypes in a model of some kind would also be helpful-ie are there subtypes which are primarily early born and others that are late born, etc?

In the MGE, there is a bias for SST^+^ interneurons being born during the early stages of neurogenesis (~E11.5-E15) while PV^+^ interneurons are born throughout the neurogenic period (~E12-P0), see Bandler…Fishell 2017 review and references therein. In the LGE, striosomal spiny projection neurons appear to be born before matrix spiny projection neurons (Kelly…Huang *Neuron* 2018). To our knowledge, almost nothing is known about the temporal birthdates of specific CGE-derived interneuron subtypes. Similarly, there are spatial domains in the MGE that preferentially give rise to specific subtypes, with a bias of SST^+^ interneurons from the dMGE and PV^+^ interneurons from the vMGE (see above and Bandler…Fishell 2017 review). We are not aware of any studies relating spatial microdomains of the LGE or CGE to specific GABAergic cell types derived from these regions.

This knowledge allows us to make some predictions about specific spatial and temporal gene patterns in relation to cell fate in the MGE (but much less so in the LGE and CGE). For example, vMGE-enriched genes such as *Bcan*, *Fabp7* and *Nr2f1* (Figures 5H and 6B) are more likely to play a role in PV^+^ fate whereas dMGE-enriched genes such as *Nkx6.2*, *Id4* and *Mpped2* (Figures 5F, 5H and 6E) may regulate production of SST^+^ cells. The downregulation of 472 genes from E12.5 MGE to E14.5 MGE is correlative for these genes being more involved in production of SST^+^ interneurons, and conversely upregulation of 522 genes over time could indicate their association with PV^+^ interneurons (log-fold change threshold > 0.25). Further studies are needed to characterize the functional role of these candidate genes in MGE-derived fate determination. And while similar spatial and temporal gene expression patterns exist in the LGE and CGE, it is nearly impossible to link these profiles to mature cell types due to our lack of understanding in this organization and lack of genetic tools to target these populations, as noted above.

4. The authors use Nestin GFP to sort out VZ cells in this study. In Figure 1 they show that Nestin- GFP cells also express markers of newborn neurons (Dcx positive), neurons (Tbr1), and IPS/neurons (Neurod6), particularly in the cortex. The authors should clarify if this is reflecting some perdurance of the GFP into post-mitotic cells? As figure 1 includes WT cells, it is unclear in subsequent figures (such as Figure 3) if they are only focusing on GFP sorted cells or if they are segregating data solely by looking at Nestin transcript. This may be a simple text fix but these missing details make the paper confusing. Due to potential perdurance of GFP protein and even transcript these are not necessarily the same.

We agree with the reviewer’s observation that the perdurance of GFP into SVZ/postmitotic cells is the most likely reason why some GFP^+^ cells may express more mature markers such as *Dcx*, *Tbr1* and *Neurod6*. The picture from the original Nestin-dVenus mouse paper (Sunabori…Okano *J. Cell Science* 2008) shows the destabilization motif significantly reduces the perdurance of GFP after the cells have left the VZ and presumably downregulated *Nestin* mRNA and Nestin protein. Their figure looks very similar to what we saw (new Figure 1A).

Even with the destabilized GFP, there will still be a delay from when the *Nestin* mRNA is downregulated vs. when the GFP protein is eliminated. Also, flow cytometry is more sensitive than immunohistochemistry to detect fluorescence, especially in live cells vs. fixed tissue. It’s likely that by sorting the cells, we are collecting cells with lower GFP levels than can be detected via immunohistochemistry, which would likely represent SVZ cells and newly born neurons. Additionally, while we always used WT controls to establish the proper gating (Sup Figure 1), we did push the GFP^+^ gate very close to the border of the WT cells to maximize harvesting both dim and strong GFP^+^ cells. Thus, we likely collected GFP^+^ cells with low levels of GFP protein that have already progressed out of the VZ/Nestin^+^ stage towards a SVZ/postmitotic fate.

Regarding the reviewer’s second point, we apologize for not being clearer in the text regarding the WT vs. Nes-dVenus cells. Cells collected from both WT and Nes-dVenus mice are included in all of the figures and analysis throughout the manuscript. Several times we note whether the cells are derived from WT or Nes-dVenus mice to highlight the enrichment of Nestin^+^/VZ cells from the Nes-dVenus mice (such as Figures2A, 7A, Sup Figure 4 and Sup Figure 12).

To clarify this point, we have added the following line in the main text, p. 5 line 109: Cells harvested from both WT and Nes-dVenus mice were used for all subsequent analysis.

5. Analysis of E14.5 is limited-especially as the authors again just sample 1 nestin GFP brain at this stage. The authors conclude that there is more diverse neurogenic and postmitotic expression at E14.5. Additional analysis of cell populations as well as validation would be important to tease out these differences and support this conclusion.

We performed 2 sequencing regions for each E14.5 GE, 1 WT and 1 Nes-dVenus, totaling ~7,100-8,650 cells per GE. As detailed in response to Reviewer #3 point #1 above, our E14.5 dataset comprises more cells and more (or equal) replicates than many recently published high-profile scRNAseq manuscripts.

To the reviewer’s second point, we have performed additional RNAscope HiPlexUP in situ hybridizations at E14.5 to (1) validate our scRNAseq data at this timepoint, and (2) confirm differential gene expression patterns between these two ages that were predicted from the scRNAseq data. Many of the genes displayed the predicted changes in gene expression patterns from the scRNAseq data, such as downregulation of *Sparc1, Sfrp1* and *Id4* in E14.5 VZ cells and upregulation of *Mir124a-1hg* and *Gucy1a1* at E14.5. This new data highlights the developmental change in transcription profiles of neural progenitors over time (specifically VZ and SVZ cells), which could have important implications in differential neuronal fates related to birthdates. These new images are now included in new Figure 8 and the corresponding text in the Results section on p. 13. We’ve also included a section in the Discussion relating these observations to the hypothesis proposed in Telley…Jabaudon 2019 that transcriptional progression of cortical APs is critical for progression of cell fate, on p. 15.

[Editors’ note: what follows is the authors’ response to the second round of review.]

The manuscript has been improved but there are some remaining issues that need to be addressed, as outlined below:The reviewers find your paper valuable and suggest several revisions that should improve the presentation of your study.Although Reviewer #1 appears satisfied with the study in the appended review, in the Reviewer Consultations offline, this reviewer agreed with the other two that a deeper analysis of the expression (in situ and immunostaining) at different developmental times (at least the 2 stages you sequenced) would verify whether a specific marker is stable or not in CGE MGE and LGE.

Figure 8 in our previous submission does directly compare in situ expression of 9 different genes between E12.5 and E14.5 in the MGE, LGE and CGE. In this case, we were focusing on genes that demonstrated different expression patterns between the 2 ages, many of which we highlighted in this Figure 8 and the corresponding last paragraph of the Results section. In this latest resubmission, we have added a Supplement figure to highlight some genes that display stable expression between E12.5 and E14.5 in the GEs.

Fabp7 characterization in particular would be welcome; if the results show selectivity, you could minimally address the request of Reviewer 4 to highlight a marker.

We don’t understand Reviewer #4’s point about *Fabp7* (see below), the scRNA-seq data and in situs are both in agreement. *Fabp7* is clearly not expressed in all VZ cells as Reviewer #4 claims, see green channel in in situ and corresponding UMAP plot in Figure 5E. We have added a Fabp7/BLBP immunostaining image below to confirm that the protein expression aligns with the in situs.

This reviewer would also like to revisit the previous request to move data concerning the dorsal progenitors to supplementary data so as to further focus on the GE, to emphasize the ganglionic eminences.Reviewer 2 asks for highlighting the impact of the study with respect to the ganglionic eminences, to better showcase the special focus of your work. This can be done by modifying the summary to better represent content of the paper.

We have modified the Title, Summary and Introduction to better highlight that our study is truly focused on the GEs per Reviewer #2’s suggestion. We don’t see in the specific Reviewer comments below about moving cortical progenitor data from Figure 1 to the supplementary data. As we stated in our previous Response to Reviewer’s, we believe that presenting all of the dorsal and ventral sequenced cells together in one introductory figure is the most comprehensive way to display our entire dataset, and thus we prefer to keep it in Figure 1. This approach allows us to compare pan-excitatory vs. inhibitory cell markers to verify the quality of the dataset (Figure 1C-D), an important initial verification to confirm the integrity of our dataset. And it’s unclear how useful it would be to segregate the dorsal cells into their own supplementary figure, as it removes the benefit of comparing/contrasting to the GE dataset. We would like to follow the common practice of snRNAseq studies, presenting the entire comprehensive dataset in the first figure and then subdividing the dataset into more refined subsets in future figures. For these reasons, we prefer to keep the dorsal cells in Figure 1 and not move extract them into a supplement figure.

Also recommended is to address in the Discussion the discordance between spatial (in situ) patterns vs sc seq for Id4 and Mest.

Our original statement of *Id4* and *Mest* as ‘divergent’ mischaracterizes these expression patterns, and we apologize for this confusion. The relevant portion of the text and figure address expression in VZ cells. Based on our in situs, *Id4* is expressed throughout the VZ of all GEs whereas *Mest* displays a more restricted expression pattern in VZ cells. There are differences, but the profiles are not divergent. However, *Id4* and *Mest* do display divergent (complementary?) expression patterns in the SVZ/MZ GE regions. This data is NOT shown in the adjacent UMAP dotplot because we’ve selected for high-*Nestin* VZ cells in this case. We have clarified our description of *Id4* and *Mest* in the text to more accurately reflect these observations.

Finally, and readily done, is to place in situs next to the relevant UMAPs in the figures; this would aid the reader's navigation through your very detailed and rich data.

We thank the reviewer for this suggestion and agree that having the UMAP plots adjacent to the in situs allows for better visual comparison between scRNA and in situ data. We have updated Figure 5, Figure 5—figure supplement 1, Figure 6 and Figure 6—figure supplement 2 so that the UMAP plots are adjacent to the in situs, with the figure legends and text references updated accordingly.

Reviewer 4 is new and asks for proof of some genes suggested to be region-specific, such as Igfbp5, and also linking progenitor heterogeneity to proliferation or fate differences or even protein levels, toward using these differences to isolate subsets by FACS. Proof that expression differences link to functional differences, by, for example, targeting candidate genes with a Cre construct, or doing localization/birthdating. Targeting genes experimentally would involve much more extensive work at this point, and Reviewer 1 especially agrees. But all three reviewers believe that because your paper is very descriptive, if you could add some functional evidence, it would increase the paper's impact. At the very least, you could try to do birthdating/cell cycle analyses, and should they work, this would help the significance of the paper. Minimally, the deeper analysis of Fabp7 expression is warranted, and should address the "ask" of Reviewer 4 to provide a marker for further cell isolation.

We discuss Reviewer #4’s concerns about these issues in more detail below.

Reviewer #3:The authors have been generally attentive to concerns raised. They have adjusted their analysis of scSEQ datasets, included new validation experiments for E14.5, and made some additional minor changes. They have revised the writing of the manuscript to explain the novelty of their findings relative to other papers (one of my original concerns) as well as to flesh out more details which increases readability. While the paper is still largely descriptive, the authors have generated some valuable new insights for the field. I do have a few remaining minor suggestions.1. The impact of this study is mainly their analysis of the ganglionic eminences where interneurons are born. In this regard the authors might consider adjusting the summary to reflect that emphasis even more. This may attract more readers to their work and increase impact.

We have modified the Title to read ‘Transcriptional heterogeneity of ventricular zone cells in the ganglionic eminences of the mouse forebrain’. And we have modified several sentences in the Summary and Introduction (highlighted in red) to emphasize that the primary focus of our study is on the ganglionic eminences.

2. The finding that Id4 and Mest spatial patterns by in situ are different than that predicted from the scseq is interesting (but also somewhat concerning for the field). It would be nice if the authors comment on this discordance in the discussion as an important point of consideration for sc seq experiments.

Our original statement of *Id4* and *Mest* as ‘divergent’ mischaracterizes these expression patterns, and we apologize for this confusion. The relevant portion of the text and Figure 5 specifically address expression in VZ cells. Based on our in situs, *Id4* is expressed throughout the VZ of all GEs whereas *Mest* displays a more restricted expression pattern in VZ cells. There are differences, but their profiles are not ‘divergent’. However, *Id4* and *Mest* do display divergent/complementary expression patterns in the SVZ/MZ GE regions. This data is NOT shown in the adjacent UMAP dotplot because we’ve selected for high-*Nestin* VZ cells in this case. We have clarified our description of *Id4* and *Mest* in the text (p. 9, lines 230-234) to accurately reflect these observations.

That being said, there is always going to be some variability between scRNA-seq and in situ datasets due to many reasons (sampling variability in scRNAseq data, different splice variants that can’t be detected with standard 3’ dropseq reactions like 10X Genomics, cell viability with scRNAseq, lack of spatial information in scRNAseq, etc.). The reviewer is correct to be somewhat skeptical of scRNAseq datasets that don’t perform some confirmation (either via in situs, immunostaining, Western blots, etc.). That’s why confirming gene expression from scRNAseq data is so critically important, and why we utilized the HiPlexUp platform to characterize 30+ genes on one brain section. The fact that the overwhelming majority of genes in our study displayed similar expression profiles between our scRNAseq and in situs instills great confidence in the accuracy of our dataset. We have added a sentence in the discussion (p. 13, lines 354-356.) noting the importance of confirming predicted gene expression patterns from scRNAseq datasets.

3. Figure 7-supplement 1F labels Nestin expressing cells. The use of red for labeling nestin is confusing here as red also labels CGE.

We have adjusted this figure, both the ‘Nestin’ and ‘Dcx’ text in panels F and G are black to eliminate any confusion with the color scheme in other panels.

4. The representation of the in situs adjacent to UMAPS in Figure 8 is really helpful. For this figure I think they should also add in annotations with the UMAPs labeled for each respective region/stage. There are labels for panel A but not the plots below. In addition, I wish the authors would use this approach in prior in situ figures-ie include the relevant UMAPs beside the in situs (for perhaps a subset), as opposed to putting in supplement. It facilitates analysis of their data.

We thank the reviewer for this suggestion and agree that having the UMAP plots adjacent to the in situs allows for better visual comparison between scRNA and in situ data. We have updated Figure 5, Figure 5—figure supplement 1, Figure 6 and Figure 6—figure supplement 2 so that the UMAP plots are now adjacent to the in situs, with the figure legends and text references updated accordingly. We hope this helps readers compare the scRNAseq and in situ data.

The dotplot in Figure 8A displays the *Nestin*-enriched population (from Figure 7E). This subpopulation clearly has distinct E12.5 and E14.5 clouds, which is why they are highlighted in that dotplot. The dotplots in Figures 8B-D contain ALL cells from E12.5 and E14.5 (from Figure 7A-D), which do not have clear, distinct E12.5 and E14.5 populations like in the *Nestin*-enriched population. That is why the E12.5 and E14.5 ovals are only shown in Figure 8A and not in Figures 8B-D.

5. Line 366 in discussion (types may be singular)

We have adjusted this line to read ‘…GABAergic neuronal cell type patterning…’.

Reviewer #4:In this manuscript the authors profile nestin-expressing cells from the 3 ganglionic eminences in E12 and E14 murine embryos by scRNA-seq. This yields several region-specific genes, some of which could serve as a new regional marker, such as Igfbp5 for CGE, but they do not prove this. Most importantly, they do not analyze the region-specific differences across at least the 2 stages they sequenced to verify if these markers are stable at least for these 2 days in development.

Figure 8 in our revised submission *did* directly compare expression profiles of 9 different genes between E12.5 and E14.5 in the MGE, LGE and CGE. In this figure, we focused on genes which had different expression patterns between the E12.5 and E14.5, which we described in detail in the last paragraph of the Results (p. 13-4, lines 332-347). To assess the temporal stability of some markers as suggested by this reviewer, we have added a new Supplementary figure (Figure 8—figure supplement 1) that depicts numerous genes with consistent expression patterns between E12.5 and E14.5. We also added a sentence in the results (p. 13, lines 342-344) referring to the new Supplementary figure.

I fully agree with the authors that it is important to explore progenitor subtypes at higher resolution, i.e. many cells, to explore heterogeneity. This is also highly relevant for the diverse output from these regions, including the generation of adult neural stem cells. However, the manuscript falls short of linking either progenitor heterogeneity or regional heterogeneity to a specific function – e.g. proliferation differences, fate differences or protein levels to allow subset isolation by FACS. Please provide at least 1 example to showcase that the expression differences link to functional differences in output (in utero-electroporation of a Cre construct targeted to a specific subset)

This question is quite similar to Reviewer #3, point #2 in our original submission, which recommended additional data linking specific genes to cell fate. Relating gene function to cell fate is an important and challenging question to tackle throughout developmental biology. We are confident that some of the genes characterized in our study likely play critical roles in cell fate, maturation and/or function, as this was a large motivation for our study.

In order to test gene function, we need the necessary genetic tools which unfortunately are not currently available for our most intriguing candidates. Our main interest is in CGE-restricted genes, as the genetic logic of CGE-derived cells is poorly understood. While the reviewer’s suggestion to introduce Cre into a targeted brain region via IUE is feasible (the Petros lab has extensive experience with this technique), this requires a mouse with the desired floxed allele to remove a gene of interest. We’ve searched the literature and databases for transgenic mice with floxed alleles for genes we are most interested in (such as *Igfbp5, Wnt7b, Rmst* and *Cxcl14*), but these do not exist. The complimentary strategy would be to have a transgenic line expressing Cre driven by genes of interest. This would allow us to fate map and characterize specific cell populations in the adult mouse, but again we could not find the desired mouse lines. Additionally, the field does not have a CGE-specific Cre-driver and thus specifically removing genes from the CGE is quite challenging (as stated in the discussion, lines 364-369). Thus, targeted gene manipulation would require extensively more work (as Reviewer #1 noted) and we don’t currently have the necessary genetic tools to perform these types of experiments. As noted in our previous Response, we hope to generate Cre-drivers for 1-2 of the most intriguing candidate genes, but this will take 6-12 months and is beyond the scope of this study.

…or highlighting by RNAscope and EdU cell cycle monitoring that the marker population has indeed specific cell cycle properties (slow division, for example).

We do not understand the reviewer’s suggested association between characterizing cell cycle dynamics with EdU and the regional expression patterns of specific genes. What does the reviewer mean by ‘marker’ population? Is the reviewer implying that certain cell populations expressing specific genes (e.g., *Igfbp5* in the CGE) have different cell cycle properties? Several studies from the 1990’s carefully explored cell cycle dynamics in the GEs and noted *slightly* different cell cycle times between different ages (E11 vs. E12) and different LGE domains (rostral, middle, caudal) in WT mice (Bhide, *JCN* 1996; Sheth & Bhide, *JCN* 1997). And there definitely are (subtle?) differences in cell cycle dynamics between VZ and SVZ cells that have been previously reported. Is this what the reviewer is referring to regarding ‘slow division’? We don’t make any claims about specific genes regulating the cell cycle properties in the manuscript, or even if specific GE subdomains have differential cell cycle dynamics. This type of EdU pulse-chase experiment could make sense if we were looking at a knockout mouse where there might be changes in cell cycle properties, but that is not the case here. Thus, we don’t understand the logic for performing cell cycle analysis in this context, and we apologize if we misunderstood what the reviewer is asking.

One specific example that raises doubts about the usefulness of the markers proposed is Fabp7, as indeed staining seems to label all cells in the ventricular zone, even though levels are always higher at boundary regions. This would mean that the "absent" expression of Fabp7 may simply be due to the depth of sequencing. Therefore, I suggest the authors perform immunostainings to verify or falsify the usefulness of this "marker".

We don’t fully understand what the reviewer is referring to regarding *Fabp7* expression. *Fabp7* is not expressed in all VZ cells based on our scRNA-seq data of the high-Nestin/VZ (new Figure 5E); note that there are clear regions of the MGE and LGE clouds where little/no *Fabp7* is expressed (less evident for CGE). Similarly, in the in situ panels (new Figure 5E green pseudocolor), *Fabp7* is only expressed in the most dorsal LGE and ventral MGE VZ cells; there is essentially no *Fabp7* expression between these border zones. In the CGE, there appears to be a smaller region where *Fabp7* is absent (in the middle CGE), which correlates with fewer CGE cells being *Fabp7­*-negative in the adjacent UMAP plots.

Thus, we believe that the scRNA-seq and in situs are largely in agreement and not in conflict. Based on this, we don’t know how to address the reviewer’s point about depth of sequencing except to say that our samples were sequenced to a sufficient depth to detect all gene expression patterns. This can be confirmed in the same image panel in Figure 5E, where *Bcan* (light blue) is expressed by a very small number of cells in the ventral MGE, which was still detected in our scRNA-seq in the lower left side of the adjacent UMAP plot.

Regardless, to convince the reviewer that the *Fabp7* expression is legit, we immunostained an E14.5 brain section for Fabp7/BLBP, Author response image 4. Note that Fabp7/BLBP protein is restricted to VZ cells in the most dorsal LGE and ventral MGE, mimicking our in situ expression pattern. We don’t believe it is worth adding this immunostained image to the manuscript because it seems odd to just have 1 immunostained image to confirm an expression pattern that is not in conflict between our scRNAseq and in situ data. But if the reviewer or editor insists, we can include this image as a supplementary figure.

**Author response image 4. sa2fig4:** 

This leads to my last suggestion namely to focus the analysis to some extent on genes encoding cell surface proteins to allow sorting subsets of progenitors.

Single cell sequencing experiments generate many candidate genes of interests, and a significant challenge is figuring out how best to follow up on these candidates. If our primary interest was to identify candidates for sorting different progenitor subsets, then focusing on cell surface proteins certainly does make sense. One could also focus on transcription factors (due to their critical role in regulating gene expression and cell fate), receptors for diffusible cues that may influence cell cycle, or many other gene categories. Rather than focusing on specific gene families, we biased our analysis to genes displaying the most intriguing spatial or temporal expression profiles (regardless of gene family/function) to highlight previously overlooked heterogeneity of these cells. We invite all researchers to download our dataset and explore the expression pattern and function of their favorite gene categories.